# INSTRUCTBRUSH: LEARNING ATTENTION-BASED VISUAL INSTRUCTION FOR IMAGE EDITING

Input IP2P+GPT-4o Ours

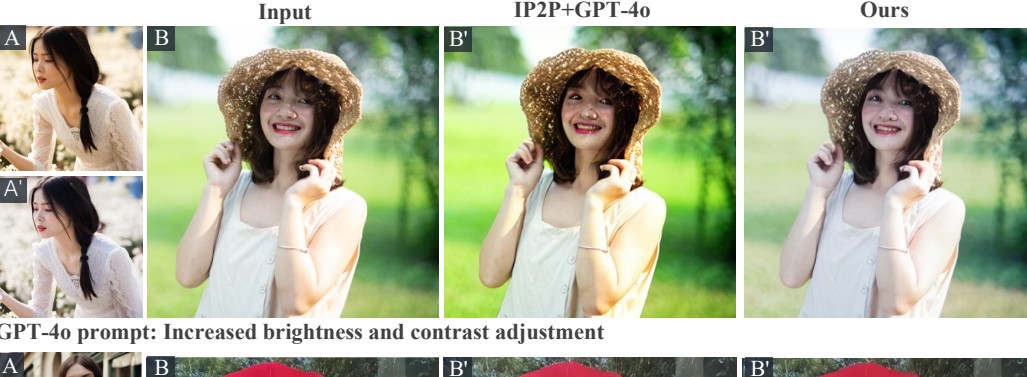

GPT-4o prompt: Increased brightness and contrast adjustment

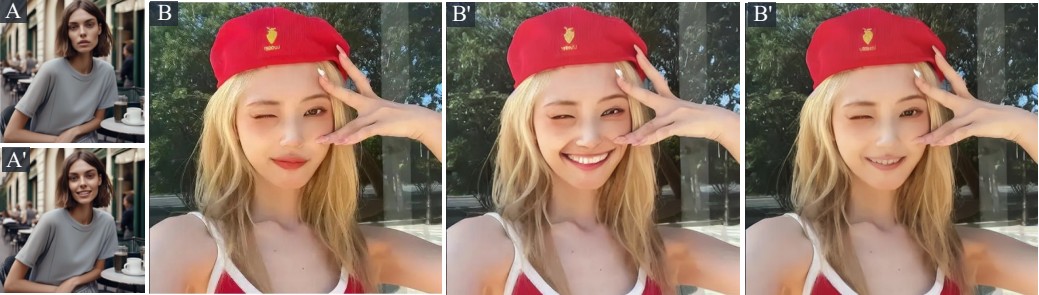

GPT-4o prompt: Subject smiling transformation

Figure 1: In terms of abstract and fine-grained edits, text-guided methods have difficulty accurately analogy the edits exhibited in reference image pairs, even with the help of multimodal large language models. In contrast, our method can better capture these edit concepts and apply them to edits of new images.

## ABSTRACT

Diffusion-based image editing methods have garnered significant attention in image editing. However, despite encompassing a wide range of editing priors, these methods are helpless when handling editing tasks that are challenging for users to accurately describe. We propose *InstructBrush*, an inversion method for instruction-based image editing methods to bridge this gap. It extracts editing effects from example image pairs as editing instructions to guide the editing of new images. Two key techniques are introduced into InstructBrush, *Attention-based Instruction Optimization* and *Transformation-oriented Instruction Initialization*, to address the limitations of the previous method in terms of inversion effects and instruction generalization. To explore the ability of visual prompt editing methods to guide image editing in open scenarios, we establish a **T**ransformation-**O**riented **P**aired **Bench**mark (**TOP-Bench**). Quantitatively and qualitatively, our approach achieves superior performance in editing and is more semantically consistent with the target editing effects. The code and benchmark will be released upon acceptance.

# 1 INTRODUCTION

Recently developed diffusion-based image editing methods Hertz et al. (2022); Tumanyan et al. (2023); Brooks et al. (2023); Xu et al. (2024) enable users to effortlessly achieve their editing goals using natural language prompts. While they have garnered significant attention owing to their flexibility and versatility in image editing, they still face challenges when dealing with editing tasks that are difficult for users to describe. Specifically, while guiding image editing with language is natural and straightforward, it becomes particularly challenging when users wish to apply analogous manipulation on others' finished edits or image transformations implemented by other tools, as shown in Figure 1. In such cases, using text or a single image as a condition to guide diffusion models for editing is quite difficult. It makes sense to provide a pair of example images to demonstrate this editing effect.

This motivates the demand for the problem of *visual prompt editing*. Similar to image analogies Jacobs et al. (2001), this problem learns an edit concept from image pairs, and subsequently applies it to edit new images. These image pairs that provide information about image transformations are also called *visual prompts*, which serves as a valuable replacement when language is imprecise in describing specific editing concepts.

One way to implement visual prompt editing is visual in-context learning Yang et al. (2024); Gu et al. (2024). It constructs the visual prompt as well as the input image and prediction noise as a grid-like input, and then uses the inpainting diffusion model to model the task as an inpainting task to predict the output. Although this paradigm can learn general image transformations by analogy with visual prompts, its performance is slightly inferior for the specific task of image editing. In addition, due to the limitation of grid input, it cannot be applied to the editing of high-resolution images. To address these issues, visual instruction inversion Nguyen et al. (2023) replaces the inpaint diffusion model with the instruction-based editing model Brooks et al. (2023); Geng et al. (2023) to improve the performance on image editing tasks while supporting high-resolution image editing. It uses the text inversion method Gal et al. (2022a) to invert the editing concepts revealed by visual prompts into the feature space of text instructions to guide the editing of new input images, but it struggles with the editing effects for two reasons: 1) Inverting instructions in textual space limits their representational ability. Since the text encoder is aligned on text-image pairs with rough descriptions, it is challenging to provide specific representations of the image editing details Chen et al. (2023d). 2) Its semantic-level instruction initialization introduces editing-irrelevant content from visual prompts, hence limiting the generalization of the instruction in generalized scenarios.

To bridge these gaps, we introduce ***InstructBrush***, an instruction inversion-based method for visual prompt editing by leveraging the instruction-based image editing model. In contrast to the previous methods, we propose the *Attention-based Instruction Optimization*. It improve the representation ability of instruction guidance by localizing and learning the editing concepts represented by visual prompts in the cross-attention layer of the diffusion model. To introduce semantic-level guidance related to editing, we introduce the *Transformation-oriented Instruction Initialization*. It ingeniously separates editing-related information from the content of visual prompts and incorporates it into the learned instructions. This effectively mitigates the risk of previous method Nguyen et al. (2023) compromising instruction generalization by introducing irrelevant content information, and promotes semantic alignment of the instruction with the objectives.

To investigate the ability of the visual prompt editing methods in guiding image editing in diverse scenarios, we establish **T**ransformation-**O**riented **P**aired **Bench**mark (**TOP-Bench**). This benchmark comprises a total of 750 images, encompassing 25 distinct editing effects, with each effect having 10 pairs of training data and 5 pairs of testing data. The creation of this benchmark not only helps to evaluate the potential of existing methods in guiding image editing, but also paves the way for further research in visual prompt editing. Qualitatively and quantitatively, our method surpasses the existing methods in terms of performance and demonstrates greater semantically consistency with the target editing effects.

In summary, our contributions are threefold:

- We introduce ***InstructBrush***, a novel solution to visual prompt editing, which extracts the editing concepts from exemplar image pairs for the subsequent image editing task.

- We propose the *Attention-based Instruction Optimization*, which is optimized within the feature space of the cross-attention, improving the representation ability of instruction guidance, and the *Transformation-oriented Instruction Initialization* to ingeniously introduce semantic-level guidance related to editing.

- We establish **T**ransformation-**O**riented **P**aired **Bench**mark (**TOP-Bench**) for visual prompt editing to assess its adaptability across diverse scenarios. Both qualitatively and quantitatively, our approach achieves more robust editing and is more semantically consistent with the target editing effects.

## 2 RELATED WORK

**Instruction-based Image Editing.** Text-guided diffusion models Nichol et al. (2021); Ramesh et al. (2022); Saharia et al. (2022); Rombach et al. (2022); Podell et al. (2023); Betker et al. (2023); Dai et al. (2023) have taken the world by storm. By leveraging the robust generative priors of these models, InstructPix2Pix (IP2P) Brooks et al. (2023) makes the initial attempt to use a triplet dataset for training a model that edits images based on instructions, achieving intuitive and user-friendly instruction-based image editing. HIVE Zhang et al. (2023b) incorporates reward learning from human feedback to fine-tune IP2P for instruction editing that is more aligned with user preferences. MagicBrush Zhang et al. (2023a) constructs a large-scale manually annotated dataset to fine-tune IP2P, greatly improving the effect in real image editing. Several existing methods, such as InstructDiffusion Geng et al. (2023) and Emu Edit Sheynin et al. (2023) extend instruction-based editing methods to new visual tasks, demonstrating its potential as a universal framework for visual tasks. Recently, some efforts Fu et al. (2023); Huang et al. (2023a) leverage Multimodal Large Language Models (MLLMs) to enhance the performance of instructions, facilitating more accurate editing. Other efforts Simsar et al. (2023); Guo & Lin (2023); Li et al. (2023a) concentration flexible and high-fidelity local editing, addressing the limitations of instruction-based editing in processing local details of images. Additionally, instruction-based image editing has been extended to 3D Chen et al. (2023b) and video Xing et al. (2023) editing tasks, showcasing its tremendous application value.

**Visual In-context Learning.** In-context learning Brown et al. (2020), which originated from the field of natural language processing (NLP), has been promoted as a learning paradigm. This paradigm enables the execution of a given task on a sample query after learning the task from a set of examples. VisualPrompting Bar et al. (2022) first introduced the concept of visual contextual learning. It uses an inpainting-based approach with grid-like inputs and has shown remarkable results in many tasks. Subsequent works Wang et al. (2023a;b); Fang et al. (2024) broaden the application areas of the framework, such as keypoint detection Wang et al. (2023a), image denoising Wang et al. (2023a), image segmentation Wang et al. (2023b) and 3D point cloud Fang et al. (2024). Recent works Wang et al. (2024); Chen et al. (2023c) introduce in-context learning on diffusion models to accomplish various visual tasks, but they require guidance from textual instructions. Yang et al. (2024); Gu et al. (2024) models visual transformations as a diffusion-based inpainting problem. However, it still requires grid images as input, which poses a significant burden when processing high-resolution images. Unlike these methods, Visii Nguyen et al. (2023) focuses on editing tasks. It inverses exemplar image pairs into a text instruction within an instruction-based image editing model, replacing textual instructions to guide the editing of new images. Our approach similarly focuses on image editing based on instruction inversion and achieves more robust editing and generalization ability to new scenarios.

## 3 PRELIMINARIES

**Latent Diffusion Models.** Stable Diffusion (SD), a variant of the latent diffusion model (LDM) Rombach et al. (2022), serves as a text-guided diffusion model. To generate high-resolution images while enhancing computational efficiency in the training process, it employs a pre-trained variational autoencoder (VAE) encoder $\mathcal{E}(\cdot)$ to map images into latent space and perform an iterative denoising process. Subsequently, the predicted images is mapping back into pixel space through the pre-trained VAE decoder $\mathcal{D}(\cdot)$. For each denoising step, the simplified optimization objective is defined as follows:

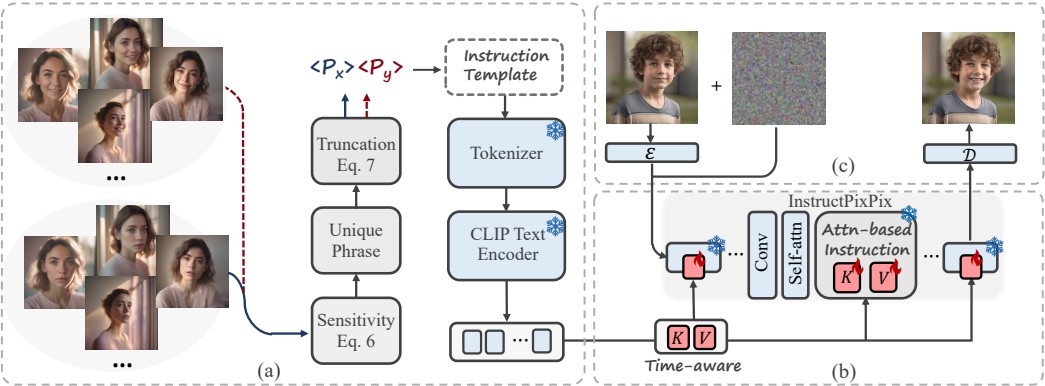

Figure 2: **The Framework of InstructBrush**. InstructBrush inverts instructions from exemplar image pairs by proposing novel (a) *Transformation-oriented Instruction Initialization* and (b) *Attention-based Instruction Optimization* modules. After optimization, the learned instructions are used to guide the editing of new images (c).

$$L_{LDM}(\theta) := \mathbb{E}_{\mathcal{E}(x),\epsilon,t}\left[\|\epsilon - \epsilon_\theta(z_t, t, \tau_\theta(c))\|_2^2\right]. \tag{1}$$

In this process, the text description $c$ is first tokenized into textual embeddings by a Tokenizer. The textual embeddings are then passed through the CLIP text encoder $\tau_\theta(\cdot)$ to obtain text conditions. The resulting text conditions are used to guide the diffusion denoising process.

**InstructPix2Pix.** InstructPix2Pix (IP2P) Brooks et al. (2023) is an instruction-guided image editing method. After encoding the input image $c_I$ using the VAE encoder, IP2P concatenates the noisy latent $z_t$ with the encoded latent $\mathcal{E}(c_I)$ in the first convolutional layer of SD. Subsequently, it uses a generated triplet dataset to perform instruction tuning Wei et al. (2021) on the improved network. This method maximizes the utilization of SD's powerful generative prior, thereby enabling stunning image editing based on human instructions $c_T$. The simplified denoising optimization objective is defined by:

$$L_{IP2P}(\theta) := \mathbb{E}_{\mathcal{E}(x),\mathcal{E}(c_I),c_T,\epsilon,t}\left[\|\epsilon - \epsilon_\theta(z_t, t, \mathcal{E}(c_I), c_T)\|_2^2\right]. \tag{2}$$

The dual conditional framework of IP2P employs both input image $I$ and text instruction $t$ for guidance, achieved through an enhanced classifier-free guidance (CFG) strategy Ho & Salimans (2022). The improved CFG incorporates two distinct guidance scales, $s_T$ and $s_I$, adjustable to balance guidance strength between text and image conditions. It learns the score estimate predicted by the network corresponding to a single denoising step as follows:

$$\begin{aligned}
\tilde{e}_\theta(z_t, c_I, c_T) = &\ e_\theta(z_t, \varnothing, \varnothing) \\
&+ s_I \cdot (e_\theta(z_t, c_I, \varnothing) - e_\theta(z_t, \varnothing, \varnothing)) \\
&+ s_T \cdot (e_\theta(z_t, c_I, c_T) - e_\theta(z_t, c_I, \varnothing)).
\end{aligned} \tag{3}$$

## 4 METHOD

The pipeline of *InstructBrush* is demonstrated in Figure 2. Based on the instruction-based image editing methods Brooks et al. (2023), *InstructBrush* inverts exemplar image pairs as editing instructions and applies them to editing new images. It proposes novel *Attention-based Instruction Optimization* and *Transformation-oriented Instruction Initialization* modules. The former introduces the editing instruction into the cross-attention layers of the instruction-based image editing model and directly optimizes the Keys and Values corresponding to the instruction within these layers, facilitating more effective instruction inversion (Section 4.1). The latter introduces semantic-level guidance related to editing, ingeniously separates editing-related information from the content of visual prompts and incorporates it into the learned instructions. This effectively promotes semantic alignment of the instruction with the objectives. (Section 4.2).

## 4.1 ATTENTION-BASED INSTRUCTION OPTIMIZATION

Inspired by Textual Inversion Gal et al. (2022a), The current instruction inversion method Nguyen et al. (2023) optimizes the embeddings of the text encoder using image pairs, aiming to represent the transformation effects between image pairs in textual space. However, the text encoder is trained on text-image pairs with rough descriptions, and its feature space is prone to losing the detailed representation of the image Chen et al. (2023d). Therefore, it is difficult to achieve the requirement of only optimizing the instruction that represents the target transformation in this space. Instead, we focus on optimizing the features in cross-attention layer of the diffusion model. These features are projected from textual embeddings to representations consistent with image features, enabling a more precise representation of image transformation details Hertz et al. (2022); Simsar et al. (2023). As a result, we introduce an attention-based instruction optimization that optimizes editing instructions in the image feature space of the cross-attention layers in the diffusion model, fostering more effective instruction inversion.

**Attention-based Instruction.** Considering a single-head cross-attention, let $Q$ be the query, $K, V$ be the keys and values from the instruction, respectively, the cross-attention is given by:

$$\text{Attention}\,(Q, K, V) = \text{Softmax}\Big(\frac{QK^T}{\sqrt{d'}}\Big)V. \tag{4}$$

Here, $K, V \in \mathbb{R}^{l \times d}$, where $l$ represents the token length of the instruction, and $d$ represents the feature dimension, the value of which depends on the position of the cross-attention layer in the U-Net framework. We optimize the features $\gamma_K, \gamma_V \in \mathbb{R}^{m \times d}$ with a length of $m \in l$ in the key and value corresponding to the first $m$ tokens of the text instruction. Because after linear projection, instruction embeddings transform from text embedding to image features, exhibiting stronger image representation capabilities. To optimize the feature embeddings of the editing instruction, our optimization objective is derived from the simplified least squares error in Eq. 2:

$$\gamma = \arg\min \mathbb{E}_{\mathcal{E}(x), \mathcal{E}(c_I), c_T, \epsilon, t} \Big[ \|\epsilon - \epsilon_\theta(z_t, t, \mathcal{E}(c_I), c_T)\|_2^2 \Big]. \tag{5}$$

Here, $\gamma = \big\{\gamma_K, \gamma_V\big\}_{1\ldots n}$ represents the features of keys and values from the first $m$ tokens of the text instruction in all $n$ cross-attention layers. The value of $m$ corresponds to the number of text tokens used for instruction initialization, as described in Section 4.2.

**Time-aware Instruction (Optional).** In the text-guided diffusion models, the denoising process focuses on image generation from low-frequency structure to high-frequency details Daras & Dimakis (2022); Zhang et al. (2023c). We believe that a similar property also exists in instruction-based editing models, where different denoising processes primarily focus on distinct transformations. We confirm this view in Figure 15. Therefore, we divide the instruction optimization equally into $j$ parts based on denoising time steps, emphasizing instruction learning within the editing-related denoising time steps. Now, we have $\gamma = \big\{\gamma_K, \gamma_V\big\}_{1\ldots n}^j$, where $j$ is 5 by default. In this way, the learned instructions can capture more details of transformations, which can guide the editing of new images more robustly.

## 4.2 TRANSFORMATION-ORIENTED INSTRUCTION INITIALIZATION

Concept inversion Gal et al. (2022a); Voynov et al. (2023); Zhang et al. (2023c) uses the semantic class word (*e.g.*, dog, cat) for initialization, providing prior information for the target concept learning. However, instruction inversion requires learning a sentence as an instruction that describes the *image transformation*. Manually initializing a sentence based on the transformation of reference image pairs is not only laborious but also subjective. The existing work Nguyen et al. (2023) utilizes the caption method Wen et al. (2023) to obtain the caption of after-editing images in the training set as the start point of the optimization. Despite the introduction of transformation-related prior knowledge, it simultaneously introduce editing-irrelevant content information about the training scenario, hindering the generalization of instruction to new scenarios. In addition, although existing multimodal large language models (MLLM) can directly compare two images to obtain a description of the differences, the daunting model size and lack of prior knowledge in professional vocabulary have caused certain obstacles in its practical application. In contrast, our approach extracts transformation-related information in a simpler and more effective way. Specifically, we first extract *unique phrases* that differentiate the images before and after editing as editing-related priors. Subsequently, we incorporate them into the *instruction template* for instruction initialization.

**Unique Phrase Extraction.** Given a set of image pairs $\{\{x\}, \{y\}\}$, where $\{x\}$ and $\{y\}$ represent the image sets before and after editing, for a single set $\{x\}$, we search for the fixed-length phrase set $P_x = \{<p_1>, \ldots, <p_r>\}$ with the highest cosine similarity between image and text features. Here, $<p_i>$ represents a text phrase from a vocabulary set, which can be customized according to the task domain or use a public vocabulary set pha (2022). And $r$ represents the adjustable number of phrases to form the caption, which is set to 5 by default. Subsequently, we compare the feature similarity between $P_x$ and the image sets $\{x\}, \{y\}$ respectively, and then measure the difference in feature similarity of the same phrase with the two sets as the *sensitivity*. This process can be represented as follows:

$$sens_i\left(<p_i>\right) = sim\left(<p_i>, \{x\}\right) - sim\left(<p_i>, \{y\}\right) \tag{6}$$

Here, $sens_i$ denotes the sensitivity of the $i$th phrase in $P_x$ and $sim$ denotes the CLIP feature similarity. We identify the phrase with maximum sensitivity as the *unique phrase* $<p_x>$ of the set $\{x\}$. However, there exist certain edits whose editing- related information cannot be recognized. To avoid the unique phrase containing editing-irrelevant information, we define the truncation conditions:

$$<p_x> = \begin{cases} <p_x> & \text{if } sens\left(<p_x>\right) \geq \eta \\ \varnothing & \text{otherwise,} \end{cases} \tag{7}$$

where $\eta$ represents a constant that controls the truncation of unique phrases, set to 0.15 by default.

**Instruction Template.** With the above method, we can get the unique phrases $<p_x>$ and $<p_y>$ for sets $\{x\}$ and $\{y\}$. Then we incorporate them into the instruction template. The form of the instruction template is strictly aligned with the base model's editing instructions to maximize the use of the textual prior. For example, we use $"turn <p_x> into <p_y>"$ as a starting point for instruction optimization. Note that when $<p_y> = \varnothing$, we use *None* instruction for initialization and optimize fixed-length features for Keys and Values in cross-attention. Although the initialized instruction is not sufficient to express the target editing effect, it can introduce prior knowledge of transformation, aiding the semantics of learned instruction to be close to the target.

## 5 Transformation-oriented Paired Benchmark

To investigate the editing capabilities of various instruction inversion methods in open scenarios and facilitate a fair comparison of these methods, We establish a benchmark named *TOP-Bench* (**T**ransformation-**O**riented **P**aired **Bench**mark), which can be utilized for both qualitative and quantitative evaluations. Our benchmark contains few-shot rather than one-shot datasets because of the effect of image transformation that is difficult to fully visualize with a single image pair. It spans 25 datasets corresponding to different editing effects. It covers a wide range of editing categories and scenarios, allowing for division from multiple dimensions. Each dataset consists of 10 pairs of training images and 5 pairs of testing images, totaling 750 images. Additionally, we provide text instructions aligned with the transformation effects for each dataset. Please refer to the Supplementary for data acquisition and detailed introduction.

To further analyze the advantages of our method, we categorize the benchmark into two different categories: TOP-Global and TOP-Local, corresponding to datasets of 14 global editing effects and 11 local editing effects, respectively. We compare the quantitative results of different methods in these two categories to validate the effectiveness of our method.

## 6 Experiments

In this section, we present qualitative and quantitative results. The implementation details of our method are detailed in Appendix C. Since the effects of image transformations are difficult to visualize completely from individual image pairs, we focus on the analysis of experiments in the few-shot setting in this chapter. Additional one-shot experiments are shown in Appendix F.1.

**Metrics**. We use several objective evaluation metrics on the benchmark. Specifically, we employ full-reference quality metrics PSNR, SSIM Wang et al. (2004), LPIPS Zhang et al. (2018), CLIP image similarity score and DINO score to assess the consistency between the generated images and the ground truth, quantifying the image editing capabilities of each method. In addition, we measure

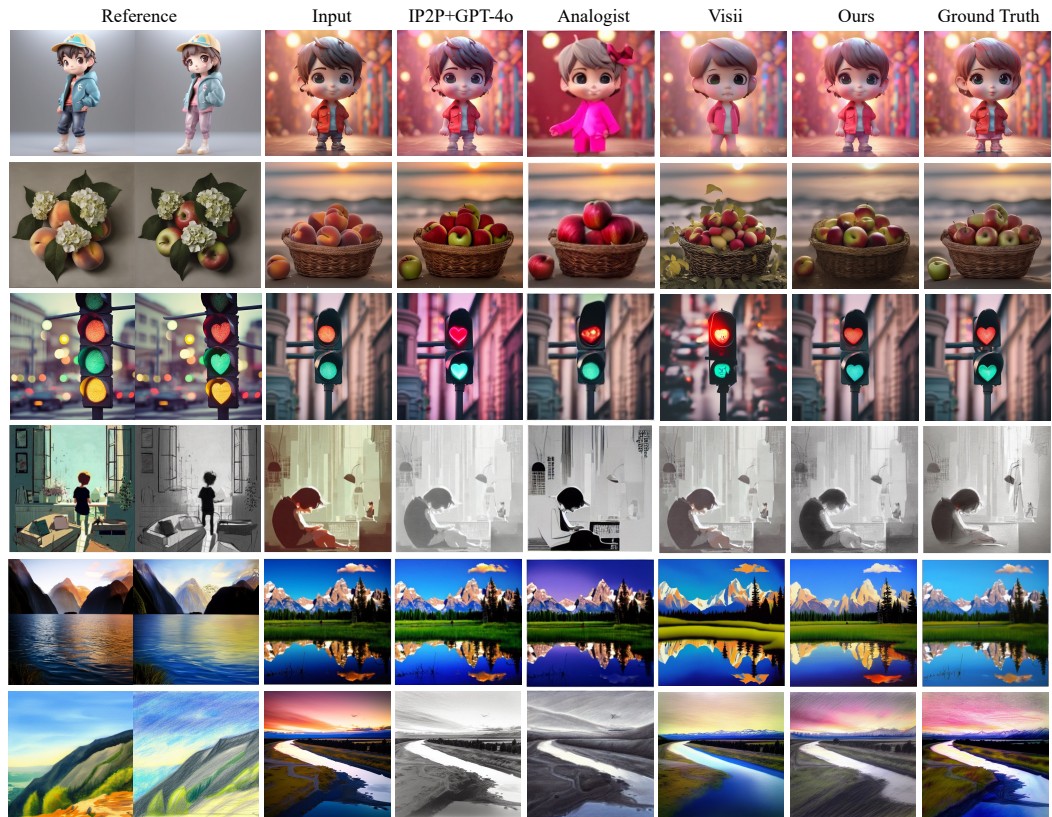

| Reference | Input | IP2P+GPT-4o | Analogist | Visii | Ours | Ground Truth |

Figure 3: **Qualitative Comparisons with Existing Methods.** Our method achieves superior performance in both local and global image editing. It effectively avoids introducing editing-irrelevant information from the training images, showing better instruction generalization.

the CLIP directional similarity Gal et al. (2022b) between image pairs to evaluate the semantic alignment between the editing direction of each method and the target. Specifically, we measure the consistency between the average editing direction from the input images to the generated images and the average direction of the training image pairs, see Appendix C for more details. Additionally, we compared the runtime of different methods, see the appendix for more details.

**Compared Methods**. We compare our *InstructBrush* with the state-of-the-art competitor Visii Nguyen et al. (2023) and Analogist Gu et al. (2024). Considering that the multimodal large language model (MLLM) can compare the differences between reference image pairs to obtain editing instructions, which can be used as the input of the text-guided editing model, we introduce GPT-4o-based IP2P Brooks et al. (2023) for comparison. We use an image resolution of $512 \times 512$ for comparison with other methods. For Visii and Analogist, we utilize its official implementation, while for IP2P, we employ its Diffusers von Platen et al. (2022) version. All experiments are conducted following the official recommended configurations.

## 6.1 COMPARISONS

**Qualitative Comparisons.** We use our *TOP-Bench* to evaluate the results of different methods. For instruction inversion methods, we employ 10 reference before-and-after editing image pairs to optimize the instruction for each editing effect. For IP2P and Analogist, we leverage GPT-4o to compare differences between reference image pairs to obtain textual captions for editing. Subsequently, we present a comparison of the results of editing the test images in Figure 3. Since the editing effects of IP2P and Analogist are mainly affected by the text conditional prior of the diffusion model rather than directly from the reference image pairs, they show certain deviations when analogizing the editing effects of the reference images. IP2P employs text instructions to guide image editing. Although such text-based editing approach cannot accurately extract the editing concepts between image pairs, the generalization ability of text and the model priors of IP2P ensure

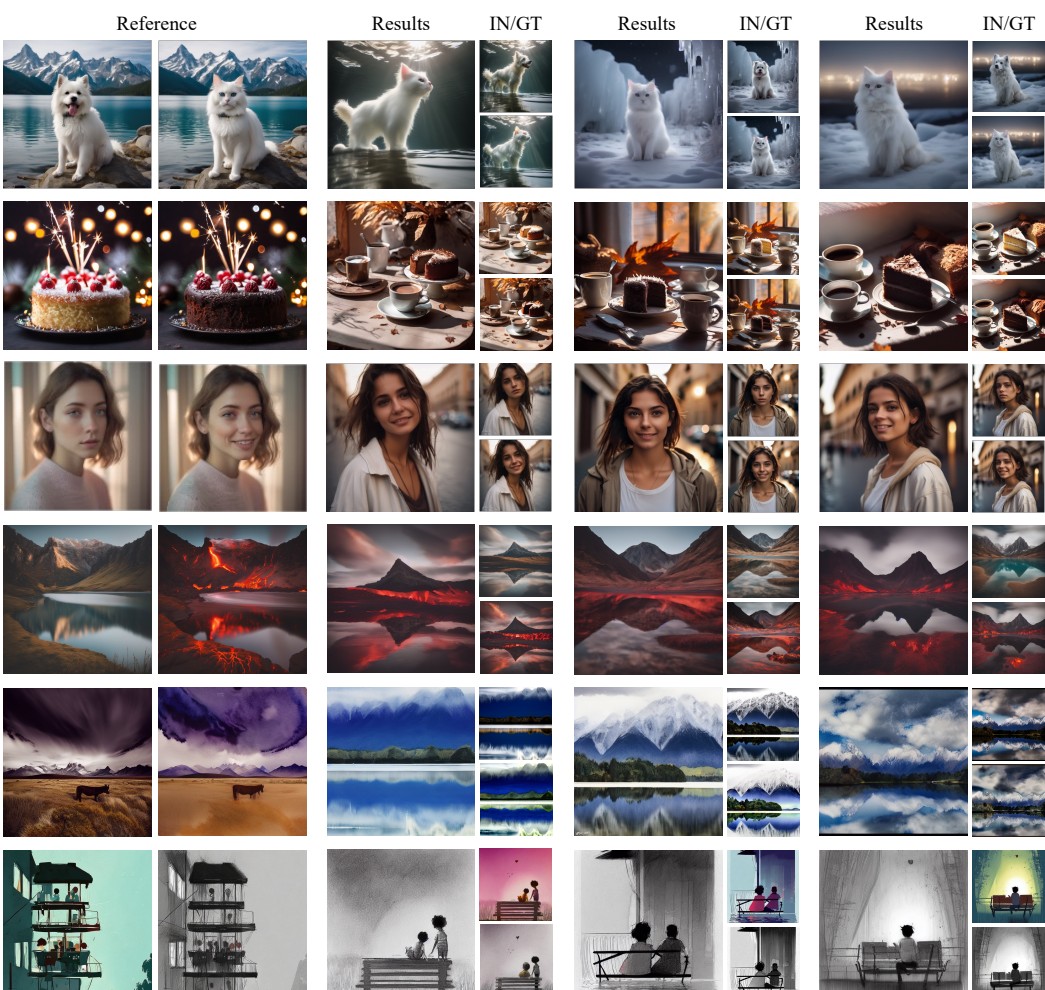

|       | Reference | Results | IN/GT | Results | IN/GT | Results | IN/GT |

Figure 4: **More Visualization Results of Our Method.** Our method demonstrates robust performance on both local and global editing. And it does not introduce scene information of the training image when editing new images, which reflects the instruction generalization of our method.

the quality of the generated images. In contrast, Analogist leverages the priors of the inpainting diffusion model, and compared to IP2P, it has a lesser understanding of the editing instructions. Additionally, the extra structural constraints imposed on the attention further exacerbate its lower adherence to the instructions. For example, suboptimal results are observed in the local edits from row 1 to row 3. Although Visii optimizes instructions to learn the target editing concept and solves the problem of IP2P not being able to specifically represent image changes using text instructions alone, its content-oriented initialization reduces the instructions generalization. It can easily introduce content information in the training image during the instruction editing process, as shown in rows 2 and 3. In addition, the limitations of optimization space also make it difficult to accurately learn target editing concepts. By contrast, our *InstructBrush* demonstrates superior editing performance. Fig. 4 illustrates more qualitative results obtained by our method, which demonstrates robust performance on both local and global editing. And it does not introduce scene information of the training image when editing new images, which reflects the instruction generalization of our method. we present additional results in Appendix F, encompassing one-shot and real-world images evaluations [1], which further substantiate the robust performance of our method.

**Quantitative Comparisons.** We conduct a detailed quantitative evaluation of these methods on TOP-Local, TOP-Global, and overall TOP-Bench. As shown in Table 1, the editing performance

---

[1]We conduct one-shot evaluations, real-world images evaluations and visualization of applications.

Table 1: **Quantitative Results.** We measure the performance of our method against several other methods based on average PSNR, SSIM, LPIPS, CLIP direction score (CLIP-D), CLIP image similarity score (CLIP-I), and DINO score. Our approach offers a basic version without the use of Time-aware Instructions as well as a complete version that utilizes them.

| Datasets | Method | PSNR ↑ | SSIM ↑ | LPIPS ↓ | CLIP-D ↓ | CLIP-I ↑ | DINO ↑ |
|---|---|---|---|---|---|---|---|
| TOP-Global | IP2P+GPT-4o Brooks et al. (2023) | 13.53 | 0.4878 | 0.3884 | 0.6897 | 0.8676 | 0.8959 |
| | Analogist Gu et al. (2024) | 12.28 | 0.3860 | 0.3993 | 0.8025 | 0.8047 | 0.8800 |
| | Visii Nguyen et al. (2023) | 15.87 | 0.4947 | 0.3866 | 0.3938 | 0.8471 | 0.8767 |
| | **Ours (basic)** | 17.51 | 0.5509 | 0.3005 | 0.2814 | 0.8541 | 0.8812 |
| | **Ours** | **18.66** | **0.5842** | **0.2526** | **0.2798** | **0.9127** | **0.9354** |
| TOP-Local | IP2P+GPT-4o Brooks et al. (2023) | 17.33 | 0.7016 | 0.2738 | 0.6188 | 0.8844 | 0.9072 |
| | Analogist Gu et al. (2024) | 14.03 | 0.5120 | 0.3334 | 0.8559 | 0.8535 | 0.9021 |
| | Visii Nguyen et al. (2023) | 18.76 | 0.7157 | 0.2585 | **0.2560** | 0.8695 | 0.9024 |
| | **Ours (basic)** | 22.36 | 0.8115 | 0.1337 | 0.3032 | 0.8887 | 0.9238 |
| | **Ours** | **23.26** | **0.8297** | **0.1143** | 0.3576 | **0.9100** | **0.9486** |
| TOP-Bench | IP2P+GPT-4o Brooks et al. (2023) | 15.20 | 0.5819 | 0.3380 | 0.6585 | 0.8750 | 0.9009 |
| | Analogist Gu et al. (2024) | 13.05 | 0.4414 | 0.3703 | 0.8260 | 0.8262 | 0.8897 |
| | Visii Nguyen et al. (2023) | 17.14 | 0.5919 | 0.3303 | 0.3332 | 0.8569 | 0.8880 |
| | **Ours (basic)** | 19.64 | 0.6656 | 0.2271 | **0.2910** | 0.8693 | 0.8999 |
| | **Ours** | **20.68** | **0.6922** | **0.1918** | 0.3140 | **0.9115** | **0.9412** |

of our method surpasses that of other methods at both the editing effects and semantic alignment. In addition, compared to the results of Visii, our method shows a more significant improvement on TOP-Local than on TOP-Global. This is because in local editing tasks, training images contain more editing-irrelevant scene information. The content-oriented initialization of Visii introduces them to the initialized instructions, posing a greater obstacle to optimization. On the contrary, our transformation-oriented instruction initialization method can accurately capture the transformations between image pairs and use them for initialization, thus improving instruction generalization.

## 6.2 ABLATION STUDIES

**Attention-based Instruction Ablation.** The use of attention-based instruction aims to avoid the limitation of CLIP space on the representation ability of target transformations and achieve a more accurate representation of image transformation details. The metrics PSNR, SSIM, and LPIPS are calculated between the output and the ground truth to evaluate the editing performance. We report results in Table 2 and observe that adopting attention-based instructions replaced with CLIP space-based instructions effectively improves the editing performance of the instructions. Additionally, we also observe in Figure 5 that compared to inversion in CLIP space, optimizing instruction in attention space has shown significant improvements in editing.

**Transformation-oriented Instruction Initialization Ablation.** Content-oriented initialization methods introduce irrelevant content information from the training images, thereby interfering with the optimization process. As depicted in Figure 5, the use of the content-oriented initialization method results in the leakage of content information from the training image into the edited image. By enabling instruction initialization to prioritize image changes over image content, it not only enhances the editing capabilities of learned instructions, but also aligns the edited image with the target transformation in terms of semantic information, which is confirmed in Table 2.

**Time-aware Instruction Ablation (Optional).** The use of time-aware instructions facilitates instruction optimization by allowing instructions to focus on learning different transformations at different denoising time steps. Table 2 explicitly shows that the use of time-aware instruction helps to improve the editing effect. The same result is confirmed in Figure 5. Note that in the second row of Figure 5, the reason the time-aware instruction does not show significant improvement is because this type of editing is relatively simple, and using other modules is sufficient to achieve such editing effects. The fine-grained facial glasses editing in the first row demonstrates the importance of this setting for fine-grained editing. Additionally, we present more qualitative ablation results in Figure 12, visualizing the importance of the module's design. However, the use of this module significantly increases the optimization time. Therefore, this module is discarded in the basic version of our method in exchange for shorter optimization time.

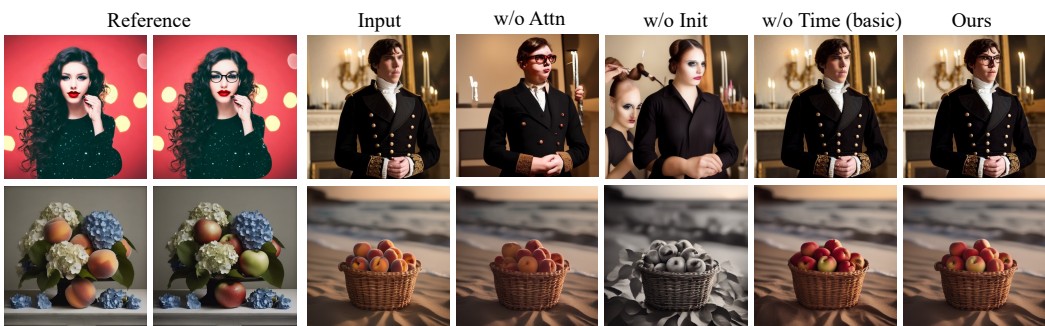

Figure 5: **Visualization Results of Ablation Study**. We visualize the independent effects of our proposed attention-based instruction, time-aware instruction, and transition-oriented instruction initialization on the results, intuitively highlighting the importance of these configurations.

Table 2: **Ablation Study**. We validate the independent impact of our proposed attention-based instruction, time-aware instruction, and transformation-oriented instruction initialization on results, emphasizing the importance of these configurations.

| Method | PSNR ↑ | SSIM ↑ | LPIPS ↓ | CLIP-D ↓ | CLIP-I ↑ | DINO ↑ |
|---|---|---|---|---|---|---|
| w/o Attn | 19.56 | 0.6709 | 0.2179 | 0.3124 | 0.8455 | 0.8757 |
| w/o Init | 20.22 | 0.6841 | 0.2018 | 0.3747 | 0.8993 | 0.9231 |
| w/o Time (Ours-fast) | 19.64 | 0.6656 | 0.2271 | **0.2910** | 0.8693 | 0.8999 |
| **Ours** | **20.68** | **0.6922** | **0.1918** | 0.3140 | **0.9115** | **0.9412** |

## 7    LIMITATIONS AND CONCLUSION

Although optimization-based methods represented by our framework are easier to learn editing concepts from multiple pairs of reference images, they increase the time cost in training. In addition, our initialization method is limited by the vocabulary used to search for unique phrases. If the phrase is not present in the vocabulary, our initialization method will initialize using *None* instruction, which will not introduce any editing prior.

Our method extracts editing effects from image pairs for editing tasks that are difficult for users to describe. It introduces a new instruction optimization and initialization method, achieving better instruction optimization and generalization. Numerous experiments have demonstrated the advantages of our method. In the future, we will apply our method to more powerful instruction-based image editing models for more robust editing performance. We hope that this work will stimulate more research and serve as a prior extraction method to aid in the training of downstream tasks.

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

## A    APPENDIX

## B    ADDITIONAL RELATED WORK

**Diffusion-based Prompt Inversion.** The diffusion-based prompt inversion methods aim to learn the text prompt from a handful of images describing concepts, thereby guiding the generation of diffusion models. Textual Inversion Gal et al. (2022a) learns text embeddings corresponding to pseudo-words to represent the target concepts. The pseudo-words can be combined with free text to guide the generation of images containing target concepts. Based on their research, some works Daras & Dimakis (2022); Voynov et al. (2023); Zhang et al. (2023c); Alaluf et al. (2023); Zhao et al. (2023) explore the effects of different inversion spaces on prompt inversion. Other works Gal et al. (2023); Wei et al. (2023); Arar et al. (2023); Chen et al. (2023a); Ye et al. (2023); Li et al. (2023b) train an image encoder based on text inversion to achieve generation guided by a given reference image. Additionally, ReVersion Huang et al. (2023b) focuses on learning the relation between objects through contrastive learning. PEZ Wen et al. (2023) inverts hard prompts by projecting learned embeddings onto adjacent interpretable word embeddings, providing a new solution for image captioning. Vinker et al. (2023) decomposes a visual concept, allowing users to explore hidden sub-concepts of the object of interest. Lego Motamed et al. (2023) uses carefully designed prompt learning methods to learn abstract concepts that are entangled with the subject from few samples. These methods focus on learning concepts to guide image generation, while our study aims to learn the transformations between image pairs to guide image editing.

## C    EXPERIMENTAL SETTINGS

**Implementation Details**. The implementation is based on one NVIDIA Tesla V100 GPU. We use public vocabulary set pha (2022) to search unique phrases for instruction initialization. Afterward, based on pre-trained IP2P Brooks et al. (2023), we optimize the features of the keys and values corresponding to approximately 10 initialization instruction tokens. Note that our method is not limited to IP2P and can also be applied to other instruction-based editing models Geng et al. (2023); Zhang et al. (2023a); Sheynin et al. (2023). We divide the learned instructions into 5 parts according to the denoising time step, and optimize each part with 1000 steps using a learning rate of 0.001 and a batch size of 1, respectively, for a total of 5000 steps. The whole training process takes about 20

minutes. During both training and inference, we adopt a text guidance scale $s_T = 7.5$ and an image guidance scale $s_I = 1.5$. And we use the Euler ancestral sampler with denoising variance schedule Karras et al. (2022) with a sampling step of $T = 20$ during the inference process.

**Evaluaion Metrics**. We use six objective evaluation metrics on the benchmark. Specifically, we employ full-reference quality metrics PSNR, SSIM Wang et al. (2004), and LPIPS Zhang et al. (2018) to assess the consistency between the generated images and the ground truth, quantifying the image editing capabilities of each method. Among them, higher PSNR indicates more similarity between the results and the ground truth; higher SSIM indicates that the results are structurally more similar to the ground truth; we implement the evaluation of LPIPS based on AlexNet Krizhevsky et al. (2012), and smaller LPIPS indicates that the results has a better features similarity between the results and the ground truth. We also use CLIP image similarity score and DINO score to assess the consistency between the generated images and the ground truth. In addition, we measure the CLIP directional similarity Gal et al. (2022b) between image pairs to evaluate the semantic alignment between the editing direction of each method and the target. Specifically, we measure the consistency between the average editing direction from the input images to the generated images and the average direction of the training image pairs. The CLIP image directional similarity Parmar et al. (2023); Nguyen et al. (2023); Gal et al. (2022b) is defined as follows:

$$1 - \cos\left(\Delta_{x \to y}, \Delta_{x' \to y'}\right), \tag{8}$$

where $\Delta_{x \to y}$ is the CLIP direction from the input image to the result image, and $\Delta_{x' \to y'}$ is the CLIP direction between the reference images.

| Num | Name | Instruction | Editing Type Local | Global |
|---|---|---|---|---|
| 1 | boy2girl | "make boy and dog into a girl and cat" | ✓ | |
| 2 | midnight | "make it nighttime" | | ✓ |
| 3 | sea painting | "turn it into a painting" | | ✓ |
| 4 | sketch style | "make the image a pencil sketch" | | ✓ |
| 5 | summer | "make it summer" | | ✓ |
| 6 | wallpaper | "make it snow" | | ✓ |
| 7 | charcoal | "turn it into a charcoal drawing" | | ✓ |
| 8 | glasses | "add a pair of glasses" | ✓ | |
| 9 | painting | "Make it a painting" | | ✓ |
| 10 | painting snow | "make it snow" | | ✓ |
| 11 | pencil sketch | "as a pencil sketch" | | ✓ |
| 12 | purple | "make the sky a deep purple" | | ✓ |
| 13 | snow | "have it snow" | | ✓ |
| 14 | watercolor | "as a watercolor painting" | | ✓ |
| 15 | 4dboy | "Turn the boy into a girl" | ✓ | |
| 16 | apple | "Turn peaches into apples" | ✓ | |
| 17 | cake | "Make it a chocolate cake" | ✓ | |
| 18 | cloud kitty | "Make the cat into a bear" | ✓ | |
| 19 | dog2cat | "Make the dog into a cat" | ✓ | |
| 20 | juice | "Make it a lemonade" | ✓ | |
| 21 | lava | "Turn it into lava" | | ✓ |
| 22 | rain | "Turn the rain into snow" | | ✓ |
| 23 | read books | "Make newspapers into books" | ✓ | |
| 24 | smile | "Add a smile" | ✓ | |
| 25 | traffic lights | "make it a heart-shaped light" | ✓ | |

Table 3: **Benchmark Presentation.** The benchmark has a total of 25 editing effects, evenly covering both local and global editing.

## D  BENCHMARK CONSTRUCTION

In recent years, there has been rapid development in text-guided image editing methods. The evaluation of image editing effectiveness has also evolved. Initially, the editing effect is solely

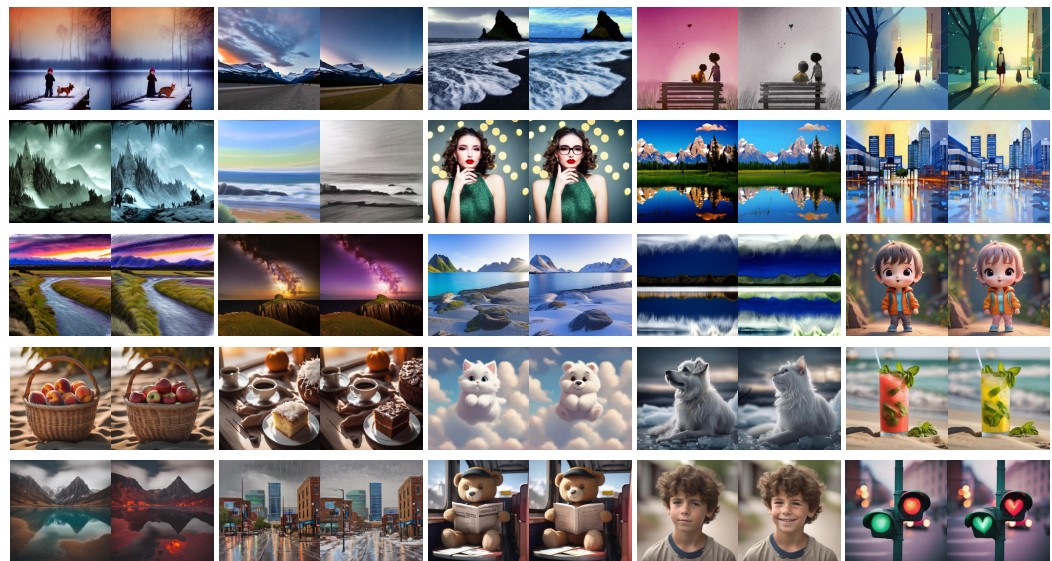

Figure 6: **Visualization of Our Benchmark.** Our benchmark spans 25 datasets corresponding to different editing effects. It covers a wide range of editing categories and scenarios, allowing for division from multiple dimensions. Each dataset consists of 10 pairs of training images and 5 pairs of testing images, totaling 750 images. We show a pair of before-and-after transformation examples for each editing effect.

evaluated through qualitative presentations and user study Hertz et al. (2022); Mokady et al. (2023), which led to significant subjectivity. Subsequently, PNP Tumanyan et al. (2023) establishes a benchmark for text-guided image editing, which assesses the performance of text-based image editing methods using text-image and image-image feature similarity scores. Later, Direct Inversion Ju et al. (2023) introduces a more robust benchmark for text-guided image editing methods, comprising 700 images and 10 editing types, and utilizes 8 evaluation metrics for an objective and comprehensive assessment. Although these benchmarks are widely used by existing text-guided image editing methods, however, the lack of paired training data prevents them from being applicable to the instruction inversion methods. Visii Nguyen et al. (2023) utilizes the filtered dataset of IP2P Brooks et al. (2023) for evaluation. However, despite being filtered by CLIP similarity, The overall quality of the IP2P training data is still poor, which is reflected in the quality and fidelity of the images before and after their editing. Furthermore, the dataset of IP2P contains fewer pairs of data for the same editing type, which hinders an accurate assessment of the performance of the instruction inversion method under the few-shot setting.

To investigate the editing capabilities of various instruction inversion methods in open scenarios and facilitate a fair comparison of these methods, We establish a benchmark named *TOP-Bench* (**T**ransformation-**O**riented **P**aired **Bench**mark), which can be utilized for both qualitative and quantitative evaluations. Our benchmark spans 25 datasets corresponding to different editing effects. It covers a wide range of editing categories and scenarios, allowing for division from multiple dimensions. Each dataset consists of 10 pairs of training images and 5 pairs of testing images, totaling 750 images. Additionally, we provide text instructions aligned with the transformation effects for each dataset.

In order to obtain paired data representing image editing, we refer to the IP2P method of generating data and utilize the existing image editing method P2P Hertz et al. (2022) to directly generate paired data before and after editing. For different editing effects, some of them completely replicate the training set of IP2P, i.e., using the image caption as well as the editing instructions are from the training set of IP2P, and the same settings of IP2P are used to generate and filter the high-quality data of the present method so as to represent the editing of the scene in the domain; while for some editing effects, we generate them through the SDXL-based P2P, while the image caption as well as the editing instructions are obtained based on GPT-4 to represent the editing of the out-of-domain scene. TOP-Bench provides paired before and after editing data. It is suitable for the evaluation of instruction inversion methods. At the same time, TOP-Bench can be segmented in multiple

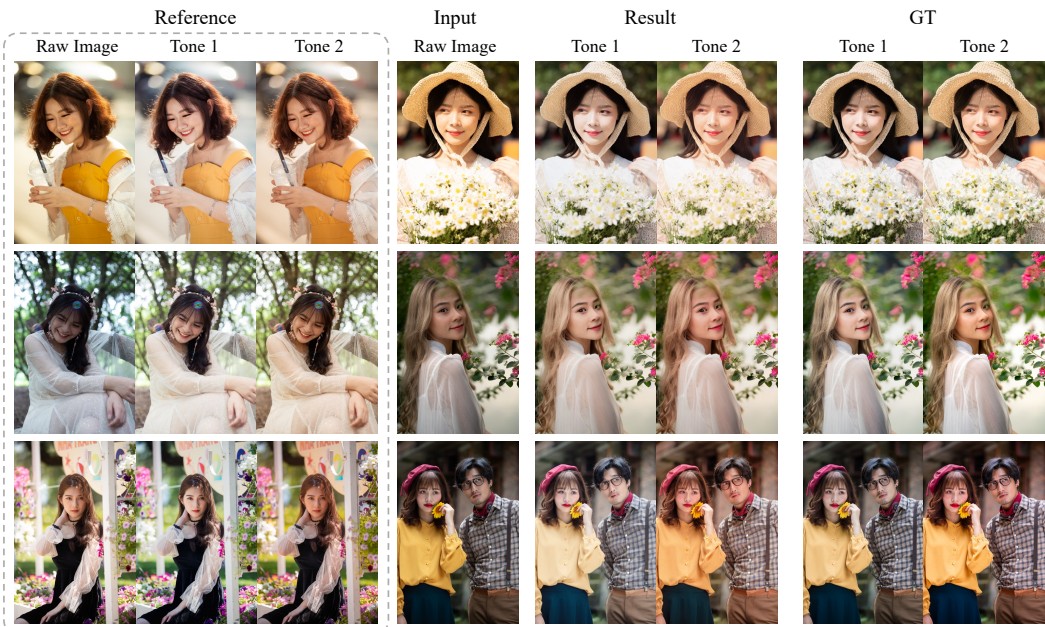

Figure 7: **Image Tone Modification.** Our InstructBrush can extract various image tones from a handful of data pairs and apply them to new images. The images, from left to right, show three reference image pairs for optimization, the input images, the corresponding editing results, and the ground truth.

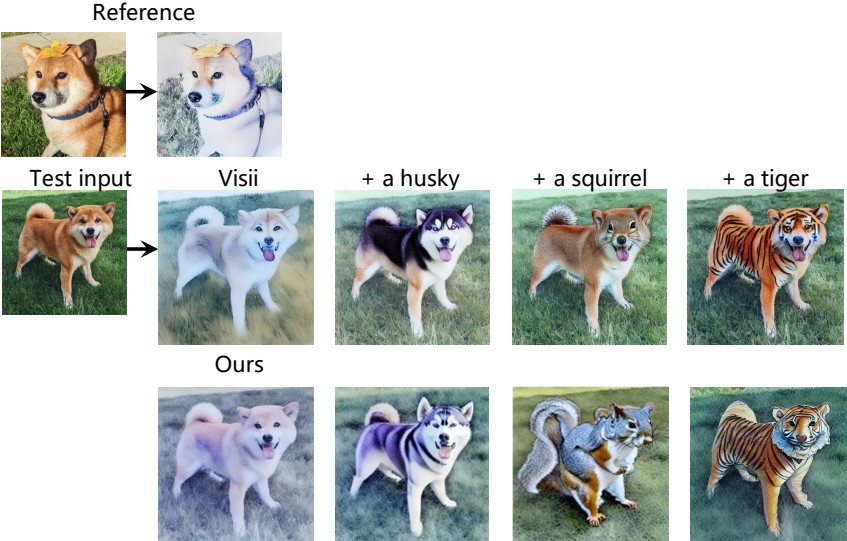

Figure 8: **Hybrid Instruction.** Our method can combine the visual instruction that represent a particular style with different textual instructions in order to jointly guide the image editing. In contrast, the visual instruction of Visii is forgotten in the process of combining with textual instructions.

dimensions to comprehensively evaluate the performance of instruction reversal methods. A detailed presentation of the datasets representing the different editing effects within TOP-Bench and their categorization is shown in Figure 6 and Table 3.

Table 4: **Extra Ablation Study**. We ablate the impact of optimizing the first $m$ tokens initialized in K, V (Ours) and the impact of optimizing all tokens.

| Setting | PSNR ↑ | SSIM ↑ | LPIPS ↓ | CLIP-D ↓ | CLIP-I ↑ | DINO ↑ |
|---|---|---|---|---|---|---|
| All tokens | 19.66 | 0.6660 | 0.2254 | 0.3214 | **0.9379** | 0.9114 |
| **Ours** | **20.68** | **0.6922** | **0.1918** | **0.3140** | 0.9115 | **0.9412** |

# E  APPLICATIONS

**Image Retouching.** Image retouching is the process of changing or improving the quality of an image. This involves enhancing colors, removing imperfections, adjusting lighting, or making other edits to improve the overall appearance of an image. Implementing image retouching using the instruction-based image editing models is challenging because the vast majority of image retouching transformations are difficult to describe using textual instructions. Our method helps in this task. Given paired data before and after image retouching, our InstructBrush can extract editing instructions representing this image transformation from the prior of generative model. The aligned instructions obtained through this process facilitate training for downstream tasks. As shown in Figure 7, our InstructBrush can extract image tones based on three image pairs that represent tonal transformations from PPR10K Liang et al. (2021) and apply these transformations to new images.

**Hybrid Instruction.** Hybrid instruction allows the use of textual and visual instructions together to guide the editing of an image. Adopting the approach mentioned in Nguyen et al. (2023), we concatenate the embeddings representing visual and textual instructions for guided editing. For visual instruction optimization, we disabled time-aware instruction optimization to achieve better hybrid instruction results. As shown in Figure, our method can combine the visual instruction that represent a particular style with different textual instructions in order to jointly guide the image editing. In contrast, the visual instruction of Visii Nguyen et al. (2023) which represents style is forgotten in the process of combining with textual instructions.

# F  ADDITIONAL EXPERIMENTS

## F.1  ONE-SHOT EDITING

To further demonstrate the advantages of our method, we test the quantitative results of different methods under 1-shot setting. We use the first pair of images from each training sets within the benchmark as our 1-shot training data pair. All settings were kept the same as in previous experiments. As shown in Table 5. Our method outperforms the other methods under 1-shot and has the same trend as the few-shot quantitative experiments. Compared to the results of Visii, our method still shows a more significant improvement on TOP-Local than on TOP-Global for the 1-shot setting. This shows in local editing tasks, training images contain more editing-irrelevant scene information. The content-oriented initialization of Visii introduces them to the initialized instructions, posing a greater obstacle to optimization. Our method can accurately capture the transformations between image pairs and use them for initialization, thus improving instruction generalization.

The 1-shot comparative experiment is shown in Figure 9. It proves that the editing effect of our method under 1-shot is better than that of Visii, while the latter easily leaks the content of the reference image pair in the result. This further verifies the advantage of the transformation-oriented initialization design of our method. Additionally, our method demonstrates more consistent editing results with reference image pairs compared to IP2P, which further demonstrates the advantage of providing image pairs for image editing. Additional 1-shot results are shown in Figure 10, which further confirms the editing effect and generalization capabilities of our method.

## F.2  EXTRA ABLATION STUDY

In Table 4, we ablate the impact of optimizing the first $m$ tokens initialized in K, V and the impact of optimizing all tokens. The results show that optimizing the first m tokens, which is our current

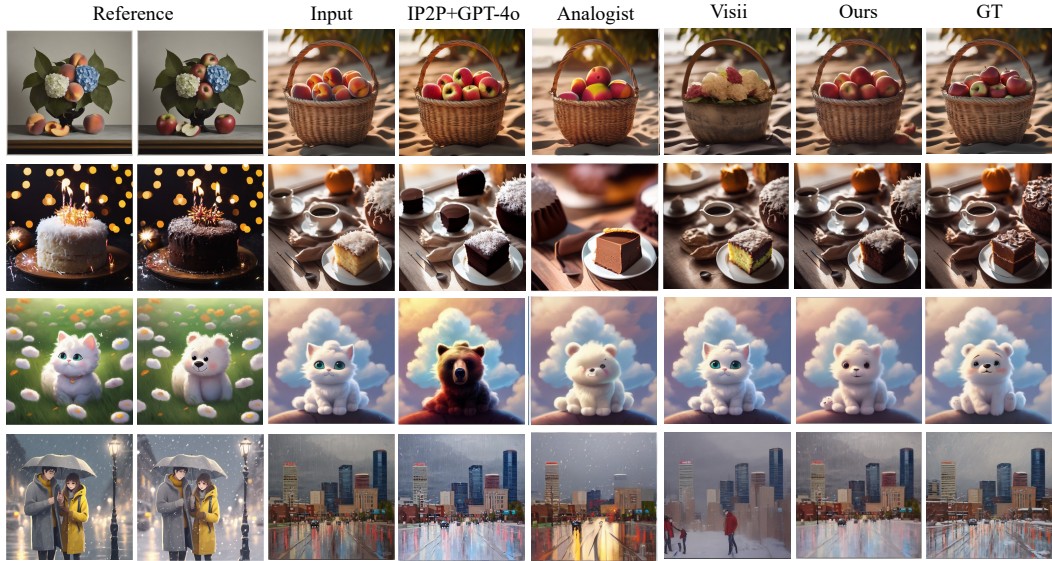

Figure 9: **One-shot Qualitative Comparisons with Existing Methods.** Our method achieves superior performance in both local and global image editing in one-shot setting. It effectively avoids introducing editing-irrelevant information from the training images, showing better instruction generalization.

Table 5: **Quantitative Results for One-shot.** We measure the average PSNR, SSIM, LPIPS, and CLIP direction scores of several methods in different editing tasks. In 1-shot settings, our method demonstrates significant superiority over other methods. We highlight in red the percentage of our method that exceeds Visii.

| Datasets | Method | PSNR ↑ | SSIM ↑ | LPIPS ↓ | CLIP-D ↓ | CLIP-I ↑ | DINO ↑ |
|---|---|---|---|---|---|---|---|
| TOP-Global | IP2P+GPT-4o Brooks et al. (2023) | 13.53 | 0.4878 | 0.3884 | 0.6897 | 0.8676 | 0.8959 |
| | Analogist Gu et al. (2024) | 12.28 | 0.3860 | 0.3993 | 0.8025 | 0.8047 | 0.8800 |
| | Visii Nguyen et al. (2023) | 16.01 | 0.5071 | 0.3692 | 0.2909 | 0.8560 | 0.8834 |
| | **Ours** | **17.79** | **0.5761** | **0.2748** | **0.3008** | **0.9009** | **0.9276** |
| TOP-Local | IP2P+GPT-4o Brooks et al. (2023) | 17.33 | 0.7016 | 0.2738 | 0.6188 | 0.8844 | 0.9072 |
| | Analogist Gu et al. (2024) | 14.03 | 0.5120 | 0.3334 | 0.8559 | 0.8535 | 0.9021 |
| | Visii Nguyen et al. (2023) | 19.73 | 0.7293 | 0.2309 | 0.5736 | 0.8794 | 0.9136 |
| | **Ours** | **23.08** | **0.8270** | **0.1172** | **0.4790** | **0.9422** | **0.9764** |
| TOP-Bench | IP2P+GPT-4o Brooks et al. (2023) | 15.20 | 0.5819 | 0.3380 | 0.6585 | 0.8750 | 0.9009 |
| | Analogist Gu et al. (2024) | 13.05 | 0.4414 | 0.3703 | 0.8260 | 0.8262 | 0.8897 |
| | Visii Nguyen et al. (2023) | 17.65 | 0.6049 | 0.3083 | 0.4153 | 0.8663 | 0.8967 |
| | **Ours** | **20.11** | **0.6865** | **0.2055** | **0.3792** | **0.9191** | **0.9491** |

method's setting, yields better effects. We attribute this to the fact that optimizing tokens with semantic information is sufficient and brings more generalization to the optimized instructions.

### F.3 MORE VISUALIZATION RESULTS

**Visualization Results Testing on Our Benchmark**. We show more visualization results of our method applied to local and global editing in Figure 16 and Figure 17.

**Visualization Results Testing on Real-world Images**. We test the performance of our method on real-world images. These data are obtained from the website as well as the PIE-Bench Ju et al. (2023). As shown in Figure 11, Our method can still achieve various editing tasks well even on real-world-images. This further validates the generalization ability of our method.

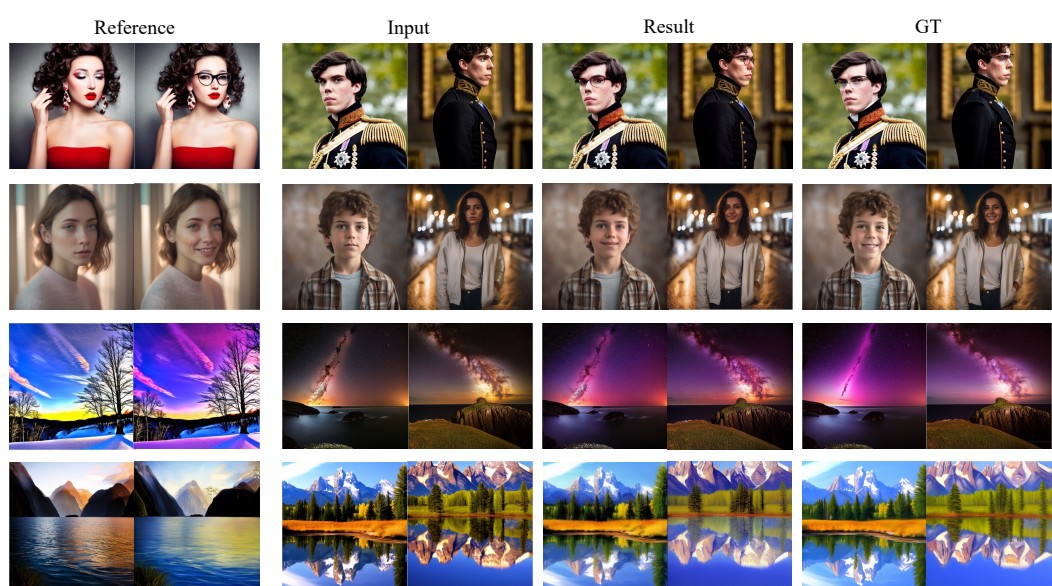

Figure 10: **One-shot Visualization Results of Our Method.** Our method demonstrates robust performance on both local and global editing in one-shot setting. Even if the training and test scenes are quite different, our method can well extract the target editing effect from the training pairs and apply it to new images. This further verifies the generalization ability of our method.

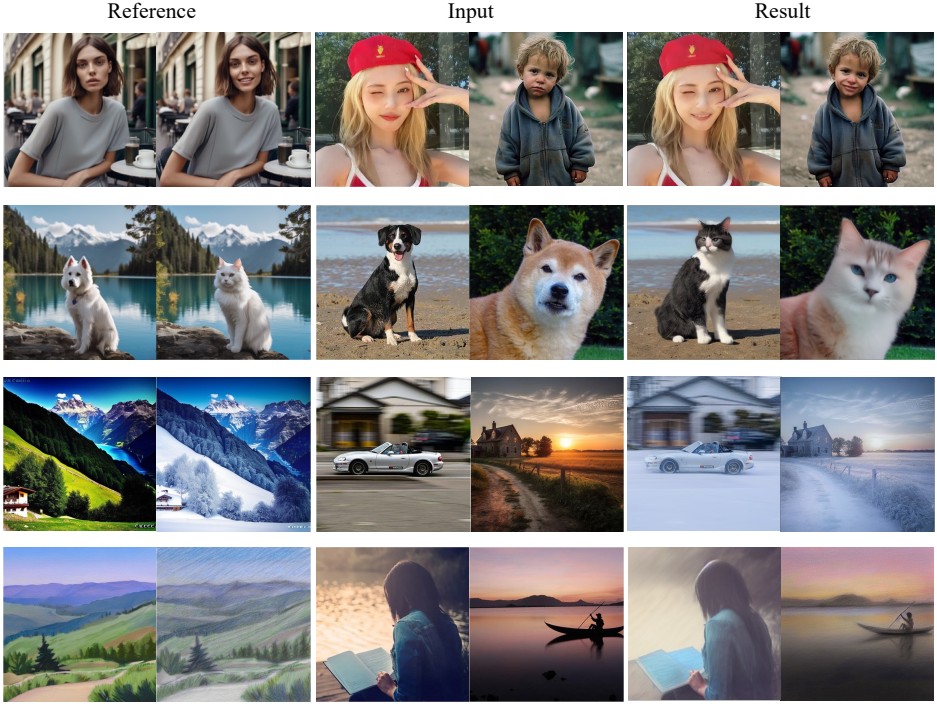

Figure 11: **Testing of our method on real-world images.** Our method demonstrates robust performance on both local and global editing. Our method also works well for editing real-world images. This further verifies the generalization ability of our method.

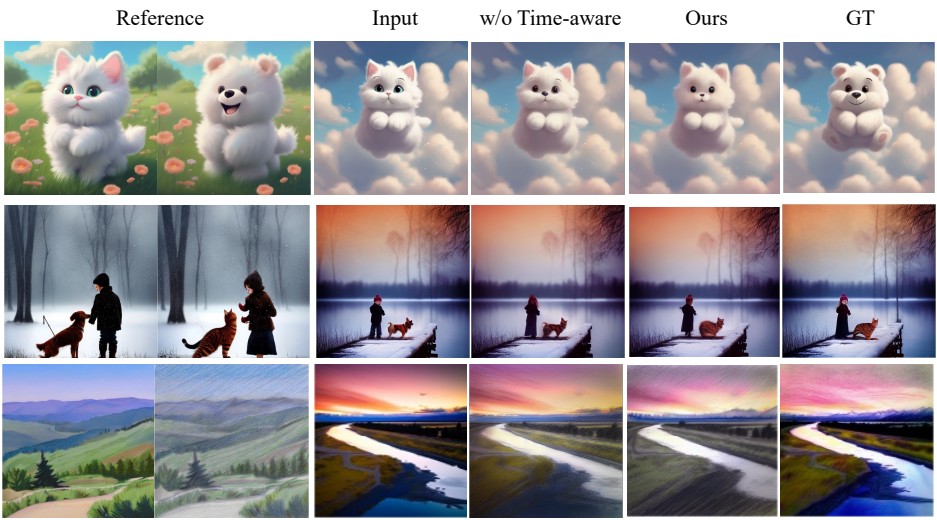

Figure 12: **Visualization effect of the Time-aware optimization method.** We provide qualitative results on the ablation of the time-aware optimization method to visualize the effectiveness of this design.

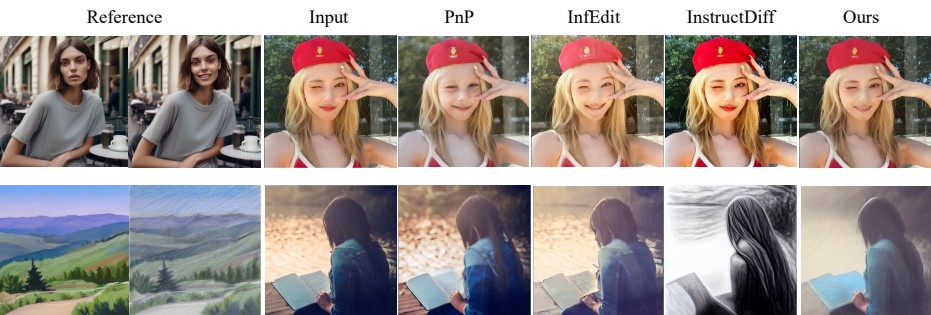

Figure 13: **Comparison with text-guided editing methods.** Our method is compared with text-guided image editing methods Tumanyan et al. (2023); Xu et al. (2024); Geng et al. (2023), and our method can more accurately compare the editing effect of the reference image pair, as shown in the figure. All text-guided editing methods use GPT-4o to obtain editing prompts.

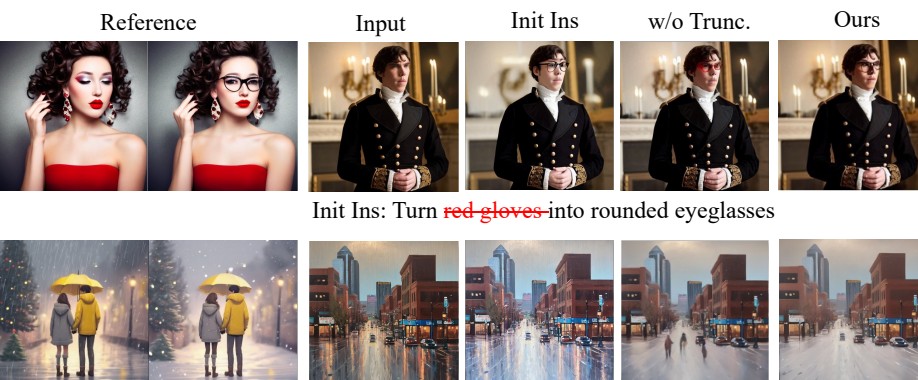

Init Ins: Turn ~~red gloves~~ into rounded eyeglasses

Init Ins: Turn ~~holding each other hands~~ into snowing

Figure 14: **Visualization of the initialization instruction.** We separately visualized the editing results of the initialization instructions obtained through our Transformation-oriented initialization method and the instructions learned through our Attention-based optimization method. Additionally, we provided the instructions before truncation to verify the effect of using truncation. Below each set of pictures, we present our initialization instructions, with the unique phrases that have been truncated indicated by red strikethroughs.

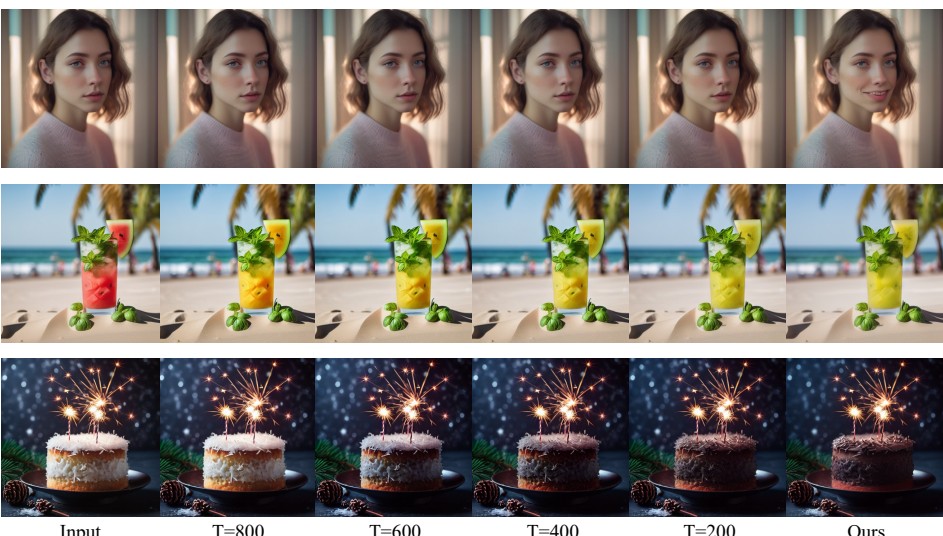

| Input | T=800 | T=600 | T=400 | T=200 | Ours |

Figure 15: **Visualization of Applying Time-aware Instructions to Various Denoising Steps.** Example: $T = 800$ represents the application of our time-aware instruction before the denoising time step of 800 (steps 1000 to 800), while the *None* instruction is applied to the denoising process after 800 steps (steps 800 to 0). Therefore, $T = 1000$ indicates the input image, and $T = 0$ indicates our full implementation. The visualization results show that in the early denoising stages, the editing focuses on coarse information such as colors (rows 2 and 3); in the later stages, the editing focuses on detailed information such as textures and facial expressions (rows 1 and 3).

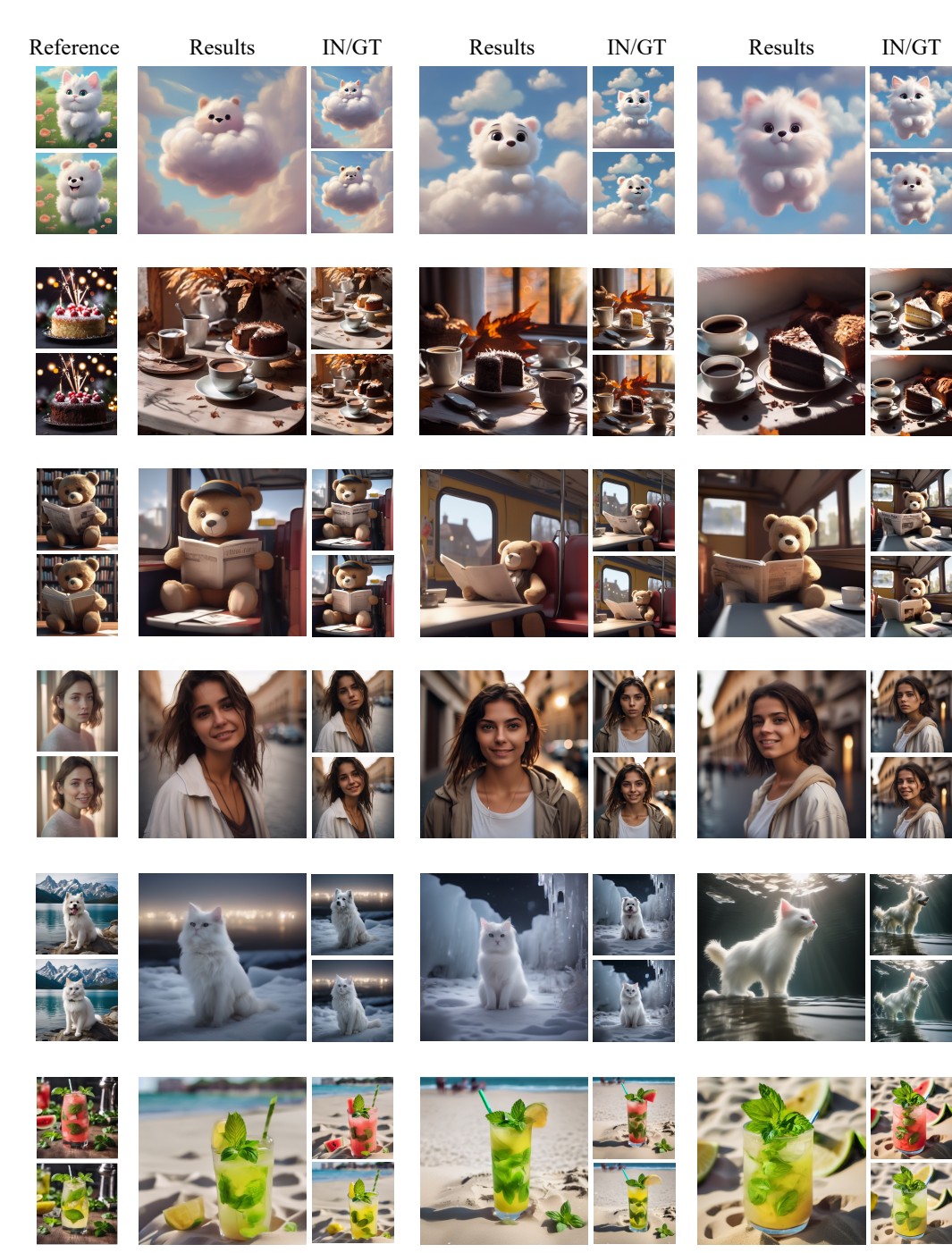

Figure 16: **More Visualization Results of Our Method for Local Editing.** Our method shows robust performance in local editing. Moreover, it does not introduce the scene information of the training image when editing a new image, which reflects the instructive generality of our method.

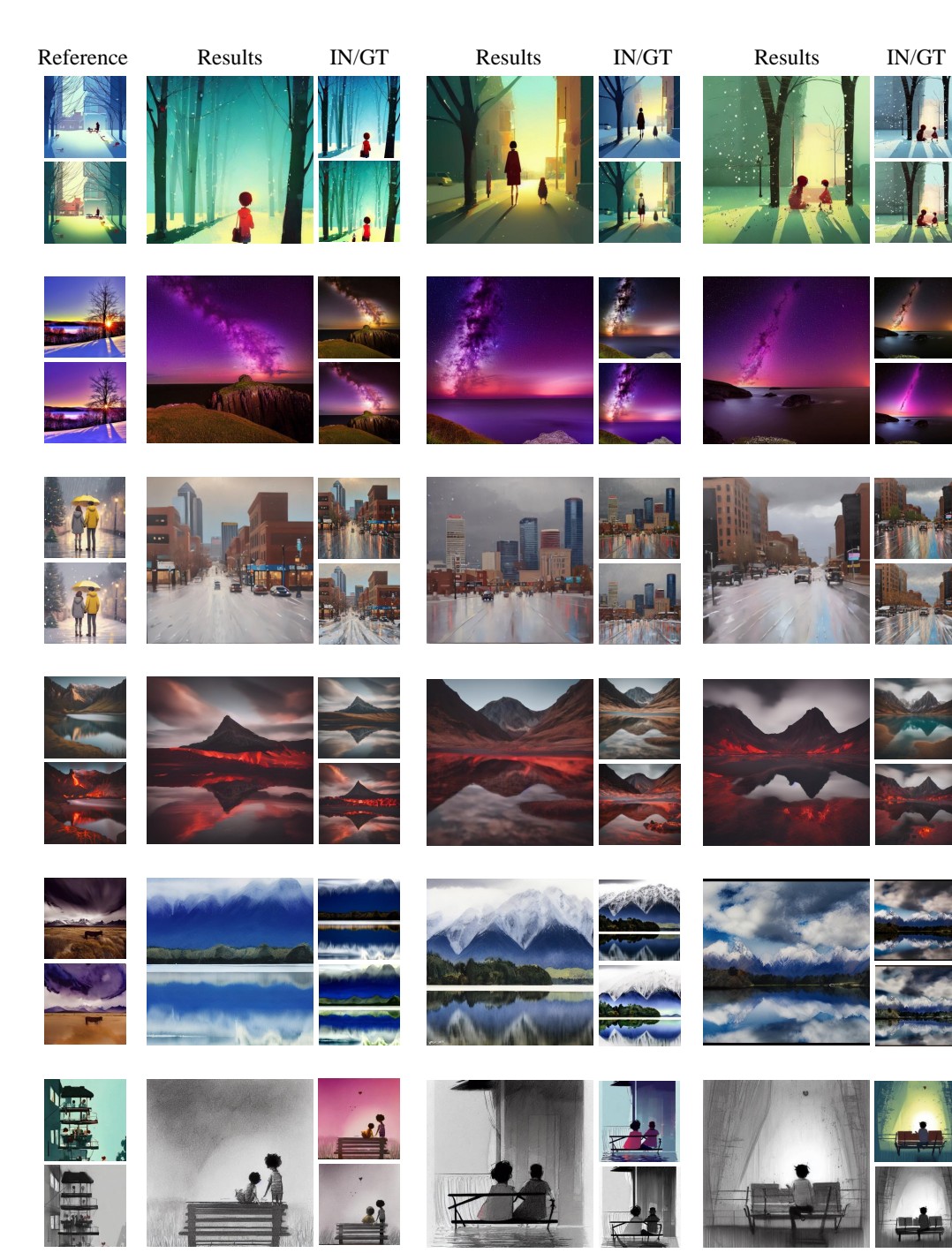

Figure 17: **More Visualization Results of Our Method for Global Editing.** Our method shows robust performance in global editing. Moreover, it does not introduce the scene information of the training image when editing a new image, which reflects the instructive generality of our method.

