# Supplementary material of InstructBrush: Learning Attention-based Visual Instruction for Image Editing

## 1 Run time comparison

Table 1: **Run time comparison.** We measure the performance of our method against several other methods based on run time.

| Method | Training | Optimization | Inference |
|---|---|---|---|
| IP2P+GPT-4o | >200hours | - | 20s |
| Analogist | - | - | 20s |
| Visii | - | 8min | 10s |
| **Ours (basic)** | - | 4min | 10s |
| **Ours** | - | 20min | 10s |

## 2 Visualization of benchmark

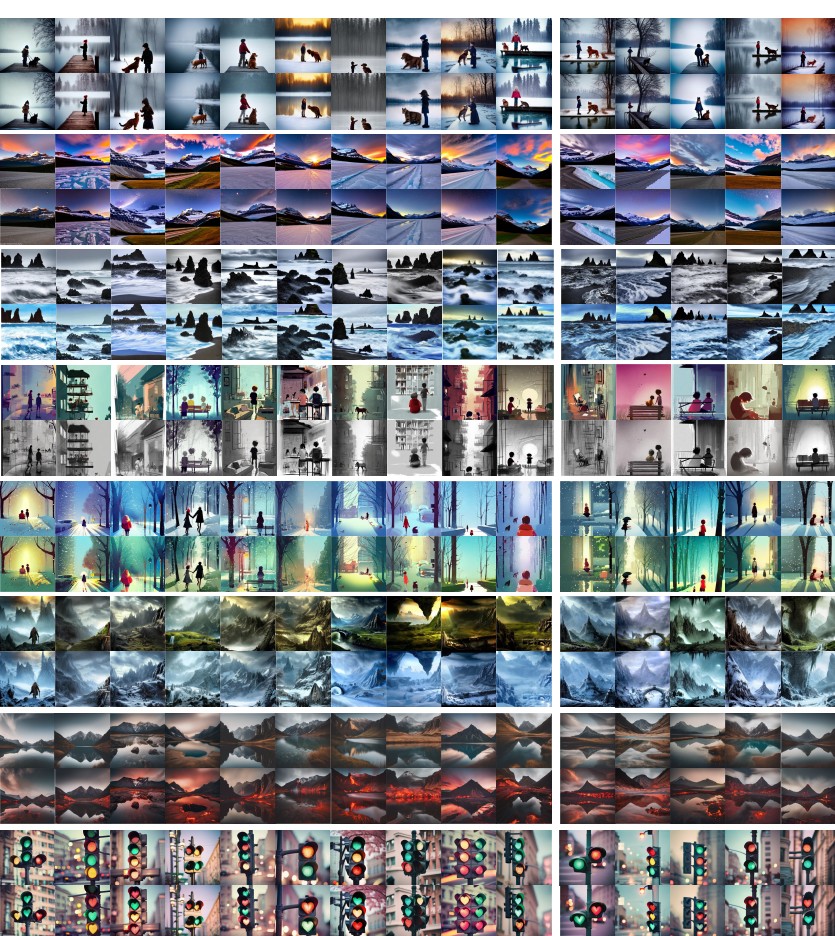

Figure 1: **TOP-Bench**. We visualize the proposed benchmark.

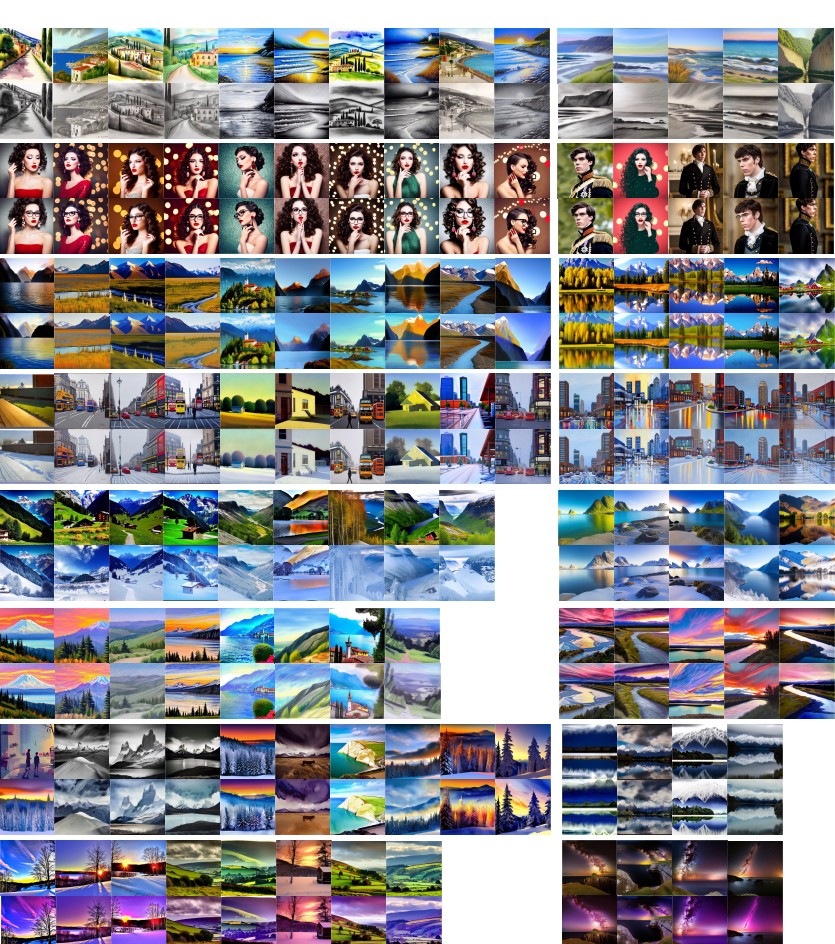

Figure 2: **TOP-Bench**. We visualize the proposed benchmark.

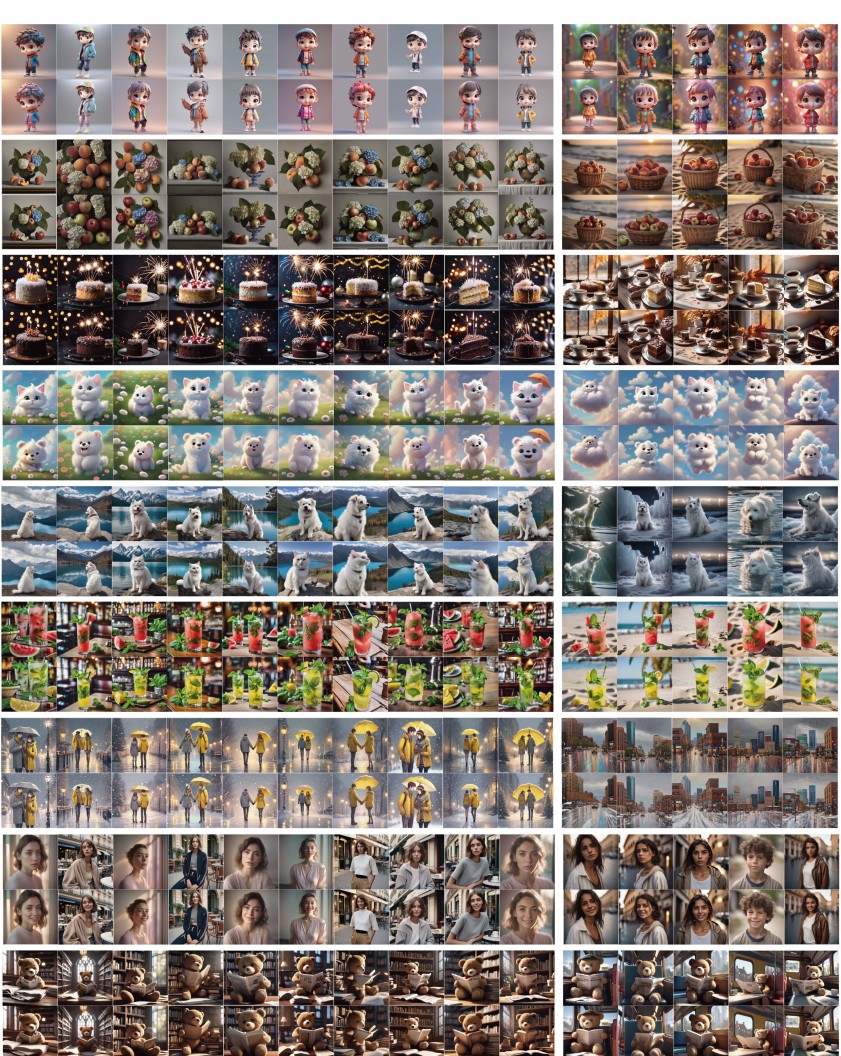

Figure 3: **TOP-Bench**. We visualize the proposed benchmark.