# OpenReview forum: "InstructBrush: Learning Attention-based Visual Instruction for Image Editing"
_ICLR.cc/2025/Conference — Submitted to ICLR 2025_

### Official Review · Reviewer_uD16 · 2024-10-16

**Soundness:** 2
**Presentation:** 3
**Contribution:** 2
**Rating:** 5
**Confidence:** 4

**Summary:**

This paper emphasizes the extraction of editing concepts from exemplar image pairs and enhances the cross-attention layers of instruction-based editing diffusion models to achieve the desired image editing results.

**Strengths:**

- The paper introduces a novel approach for visual prompt editing and proposes a well-justified generation strategy.
- The proposed method yields more satisfactory results compared to other approaches on the given task.
- The writing is clear and easy to understand.

**Weaknesses:**

- Input Constraint: A key issue is the requirement that the differences between the reference image pairs used for editing must not be too large. Since the model relies on Eq. 6 and Eq. 7 to extract differences at the pixel level, acquiring such image pairs is challenging. A possible way to obtain these pairs would be through text editing methods or the prompt-to-prompt approach. However, this makes the input conditions much more complex compared to simply providing text, significantly reducing the practicality of the proposed task design.

- Redundancy of Exemplar Image Pairs: The paper employs Unique Phrase Extraction to derive concepts from the image pairs and then uses the extracted textual features for editing. This raises questions about the necessity of using exemplar image pairs as input. If textual descriptions were provided directly, similar results could be achieved. The paper includes experiments with GPT-4v + IP2P, which demonstrate the effectiveness of this approach to some extent. However, I believe that slight modifications to GPT-4v + IP2P could achieve results similar to those proposed in the paper, deepening my concerns about the necessity of the method.

- Lack of Novelty: The Time-aware Instruction strategy used in the paper is too common, and the core idea resembles methods like recaptioning and null-text inversion. While the novelty issue is not as critical as the previous two concerns, it still raises doubts about the paper's originality.

**Questions:**

My main concerns are outlined in the Weaknesses section. While I acknowledge the authors' efforts in producing promising results, both the necessity of the task and the design of the method leave too many points that need improvement and discussion. Therefore, I believe the paper is not ready for publication.

---

> ### Author Response · Authors · 2024-11-20
>
> Thank you very much for your constructive comments. We addressed you comments and questions and revised the paper. We summarize the answers below.
>
> **[W1]  Input Constraint**
>
> We respect but disagree with your view regarding the proposed method's setup. Essentially, our method is not intended to enhance the effectiveness of general image editing tasks, but rather aims to accurately replicate the transformations between images pairs and apply them to new images. This setup has been thoroughly explored in the 'image analogy' task [1][2]. It uses pairs of images before-after transformation as conditions and provides an input image as a query to analogize to the result. Therefore, providing example image pairs to represent editing effects is necessary. Moreover, some existing works have explored representing transformations through example image pairs (image prompts) to guide image editing [3][4][5], thus the setup of this study has a solid theoretical background.
>
> Moreover, because the example image pairs are responsible for demonstrating the transformation effects, they naturally contain similar parts. If the two images differ too much, it may be challenging to intuitively reflect the transformation concepts involved. Regarding the practicality of our method, we have tested various types of transformation effects. In addition to the editing covered in TOP-Bench, such as global editing, local editing, multi-object editing, shape editing, style editing, identity replacement, object replacement, and face attribute manipulation, we have also introduced image tone modification (*Figure 7*), hybrid editing (*Figure 8*), and real-world image editing (*Figure 11*). The results indicate that the method has strong generalization capabilities.
>
>
>
>
>
> **[W2] Redundancy of Exemplar Image Pairs**
>
> For the reviewer's concern about the necessity of using image pairs as input, we have already explained this in detail in **[w1]**.
>
> Furthermore, it is necessary for us to reiterate the implementation process of our method. The primary purpose of using image pairs as input is to serve as a supervisory signal, aiding the feature optimization process of the editing instructions. The role of using image pairs to acquire unique phrases is to introduce semantic information into our optimization process. Therefore, our method is essentially based on optimization to learn image pair-based editing, similar to the image analogy task [1][2]. This optimization approach can directly learn pixel-level or even mask-level transformations between image pairs, which are difficult for text-guided editing methods like GPT-4+IP2P to capture.
>
> While it might be possible to introduce semantic priors into our optimization process through slight modifications to GPT-4, our unique phrase extraction method is more lightweight and efficient. In addition, it supports customization of the phrase dictionary and CLIP models to better suit downstream tasks.
>
>
>
> **[W3] Lack of Novelty**
>
> We fully understand the reviewer's emphasis on innovation. but as *Reviewer 8Vpr* stated in *Strength 2*, we emphasize that the innovation of our method does not lie in proposing a novel module. Instead, it delves into the extraction of editing concepts based on image prompts. We leverage the prior implementation of Instruction-based image editing models and innovatively introduce the KV token optimization in Attention for image prompts learning to represent the target editing instructions, enhancing the representational power of the concept. Furthermore, to further explore the impact of instructions on editing process, we introduce a time-aware method to optimize the instructions at different denoising times, comprehensively investigate the representation of editing instructions, and provide a clear solution for visual prompt editing.
>
> [1] Hertzmann et al., *Image analogies*, SIGGRAPH 2001.
>
> [2] Liao et al., *Visual attribute transfer through deep image analogy*, SIGGRAPH 2017.
>
> [3] Nguyen et al., *Visual instruction inversion: Image editing via image prompting*, NIPS 2023.
>
> [4] Yang et al., *Imagebrush: Learning visual in-context instructions for exemplar-based image manipulation*, NIPS 2023.
>
> [5] Gu et al., Analogist: *Out-of-the-box Visual In-Context Learning with Image Diffusion Model*, SIGGRAPH 2024.

---

> > ### Comment · Reviewer_8Vpr · 2024-11-21
> > **A comment on the visual prompt setting**
> >
> > As I read other reviews, I will let the authors address each reviewer's concerns.
> >
> > However, I want to support the authors on the question regarding "Input Constraint." Visual Prompting for Image Editing or Image Analogies has been extensively studied in Image Editing/Image Manipulation (as referenced by the authors), and instead of considering it an "input constraint," I believe that "visual prompting" might actually clarify the edits, especially in cases where text alone might be ambiguous.
> >
> > ---
> >
> > For example, an edit such as "Turn it into a painting" might be interpreted in many different styles of painting (e.g., watercolor, impressionist, pencil sketch). Even a more refined edit like "Turn it into a watercolor painting" might still allow for different interpretations of the color scheme (e.g., pastel, pink, dark tones).
> >
> > Thus, there are many cases where it is far easier to convey the desired edit through a {before, after} image pair. For instance, in the Teaser Figure, the edit is described as "Increased brightness and contrast adjustment." In this case, a user might edit the first photo and produce the {before, after} edits. It could be difficult to describe exactly how much brightness and contrast the user wants (as it often depends on visual judgment). In such scenarios, it would be more practical and intuitive to provide the system with {before, after} images and ask it to transfer the same changes to a new test image.
> >
> > That said, I agree that in the current paper, the authors could do a better job of illustrating these scenarios (e.g., cases where edits are difficult to describe using text alone). For references supporting this point, "Diffusion Image Analogies" (SIGGRAPH 2023) and "Visual Instruction Inversion: Image Editing via Visual Prompting" (NeurIPS 2023) include several relevant examples.
> >
> > ---
> >
> > Lastly, it is worth mentioning that using visual prompting does not mean completely replacing text prompts. In fact, the two approaches can complement each other. For example, Figure 8 demonstrates combining a text prompt such as "Turn a photo of a dog into watercolor style" with {before, after} images, while also transforming the dog into a "tiger" using text prompts.
> >
> > Therefore, I believe that the "Input Constraint" point should not be considered a weakness of this paper.
> >
> > ---
> >
> > For other concerns, I have no comments and will leave it to the reviewers to judge. Thank you!

---

> > > ### Comment · Reviewer_uD16 · 2024-11-22
> > > **Response to Reviewer 8Vpr**
> > >
> > > I’d like to clarify my response further. My main point is that the input setting in this paper has issues. To be clear, I’m not questioning the value of using image prompts as a research direction—quite the contrary. It is one of the hottest areas of research right now, with many outstanding works such as IP-Adapter and OmniGen making significant contributions. However, the problem lies specifically with the input setup in this paper. The reference image pairs shown in the paper are clearly obtained through image editing, making them difficult to acquire in practice. What practical application scenarios could realistically use such an input condition?
> > >
> > > Regarding the example you mentioned, “Turn it into a painting,” is it easier for users to provide a textual description (e.g., watercolor, impressionist, pencil sketch) or to go out of their way to obtain the type of image pair used in this paper? If extracting specific information from images for editing is the goal, methods like LoRA, DreamBooth, and StyleDrop are more practical and efficient alternatives, not to mention recent advancements like IC-LoRA.
> > >
> > > Leaving aside the task setup, the methodology in this paper also lacks novelty. The first stage could be entirely replaced by Multimodal Large Language Models, which would analyze the input far more effectively than the phrase set-based feature matching used in this work. As for the second stage, the improvements in image generation at the cross-attention layer show little innovation. I genuinely fail to see what is particularly groundbreaking about this paper.
> > >
> > > Additionally, your statement, “I will let the authors address each reviewer’s concerns,” **gives the impression that you might have a connection with the authors**. If that is the case, I’d like to ask: by insisting on covering up the issues in this paper and pushing it to acceptance, wouldn’t that invite skepticism from other researchers and potentially harm the authors’ reputation in the long run? In my opinion, this paper cannot meet the acceptance standard simply by supplementing additional experiments. Therefore, I apologize, but I stand by my opinion of “borderline reject.”
> > >
> > > Since I have no power to decide whether the paper is ultimately accepted, perhaps you could direct your persuasion to the AC rather than to me. That said, I do not plan to spend any more time on this paper. Thank you for your understanding.

---

> > > > ### Comment · Reviewer_8Vpr · 2024-11-22
> > > >
> > > > Hi Reviewer uD16,
> > > > First of all, I respect your opinion and your evaluation of this paper. That's why I wrote "I will let the authors address each reviewer’s concerns... I have no comments and will leave it to the reviewers to judge."
> > > >
> > > > But for the point of Input Constraints, I just simply want to write to support authors about problem settings (Visual Prompting), and hopefully can spark a healthy research discussion. I keep my opinion that the problem setting is valid, and in my opinion, should not be the weakness of the paper. I've made it clear that I do not have comment on other issues that you've pointed out for this paper -- And if you rate this paper below acceptance bar for other reason, I respect your decision and judgement. That's why I wrote "I have no comments and will leave it to the reviewers to judge.".
> > > >
> > > > ---
> > > >
> > > > Please do not assume anything, and indeed, *you assume that I have relationship with the authors*. This statement, even in bold, might harm my reputation and authors' reputation. We can let Area Chairs and Program Chairs check this.
> > > >
> > > > Again, **I do NOT know any authors of this paper, and I do NOT have any connections to the authors**.
> > > >
> > > > ---
> > > > I simply want to spark a healthy research discussion. I am so sorry for any confusion this might causes. But in the future, please be careful with these assumption like this.

---

> > > > > ### Comment · Area_Chair_m1KZ · 2024-11-22
> > > > >
> > > > > Take it easy, guys.
> > > > >
> > > > > I see nothing that suggests 8Vpr has a connection to the authors.
> > > > >
> > > > > This seems like a misunderstanding. Let's try to have a civil discussion and focus on the paper.

---

> > > > > > ### Comment · Reviewer_uD16 · 2024-11-23
> > > > > > **Response to Area Chair m1KZ and Reviewer 8Vpr**
> > > > > >
> > > > > > Thank you to the Area Chair and Reviewer 8Vpr for your responses. I apologize for questioning the relationship between Reviewer 8Vpr and the authors. Perhaps I didn’t express myself clearly—I would not assume that a paper should be rejected just because such a relationship might exist. I only raised the question because I found your responses somewhat puzzling. Moreover, if my intention were to push for a rejection, I would have rated it as a 3 from the start.
> > > > > >
> > > > > > I am simply fulfilling my responsibilities as a reviewer by providing my opinions. I genuinely believe that this paper cannot be accepted solely through additional revisions, particularly due to the task setup. Using such similar input images is both highly challenging and uncommon, which I think is the most critical issue. Additionally, the novelty does not seem to reach the level of a borderline acceptance. Of course, these are just my opinions for reference. Regardless of whether the paper is accepted or not, I will respect and support the final decision.

---

> > > > > > > ### Author Response · Authors · 2024-11-23
> > > > > > >
> > > > > > > We sincerely thank all the reviewers for taking the time to read our paper and for providing constructive feedback. We believe that, regardless of whether this paper will be accepted in the future, it will help us to improve the quality of our paper and deepen our understanding of all related issues. We are grateful once again for the reviewers' patient responses.

---

> > > > > ### Author Response · Authors · 2024-11-23
> > > > > **Author's Response to Reviewer 8Vpr**
> > > > >
> > > > > We sincerely thank you again for your support, and we are very sorry for any trouble this matter has caused you.

---

> > ### Comment · Reviewer_uD16 · 2024-11-22
> > **Response to authors**
> >
> > First, I sincerely thank the authors for taking the time to respond to my comments. Below are my further replies.
> >
> > ### Input Constraint
> >
> > Regarding the authors’ reference to the "image analogy" task, the two cited papers are from 2001 and 2017, predating the significant breakthroughs brought by diffusion models. At the time, this task was indeed interesting. However, since 2022, diffusion models have demonstrated their capability to achieve high-quality user-friendly-condition-to-image generation, rendering tasks like the "image analogy" task solvable through alternative strategies.
> >
> > Even putting this aside, obtaining the kind of minimally different image pairs required by your method is inherently challenging. One would either need to use instruction-based editing models or other textual-based editing methods to generate the image pairs, or extract frames from videos. For the first case, if users already have such a powerful textual editing model, users could achieve the desired result using the original prompts or fine-tuning, without resorting to the cumbersome process of acquiring image pairs. For the second case, your method does not seem to offer any significant advantages over animation editing approaches, such as using flow-based warping techniques.
> >
> > Even in extreme cases where an image pair is indispensable for achieving the desired effect and textual-based editing models fail, the likelihood of encountering such scenarios is quite low. Moreover, based on the visual results presented in the paper, I don’t believe your method is essential. Results generated by IP2P + GPT are perfectly acceptable. Additionally, since your method’s improvements are primarily focused on the text input while the diffusion model still relies on textual guidance, its stability and generalization capability remain questionable.
> >
> > In summary, given the current progress in image generation research, this task seems more like a "for-publication" task with limited practical applications.
> >
> > ### Novelty
> >
> > Even if we disregard the task setup, the two core contributions of this paper, put simply, are:
> >
> > - Extracting phrases from image pairs, which is a task that can already be effectively accomplished by vision-language large models. Many studies have explored this direction and achieved significant results. I suggest the authors delve deeper into more cutting-edge work, particularly research combining MLLMs with image generation or recaptioning.
> >
> > - Modifying the cross-attention layer features based on diffusion model timesteps. This strategy borrows heavily from numerous existing methods, and I don’t find it to be an original innovation. Similar improvements have been extensively explored in prior work.
> >
> > Overall, for a paper to be worth acceptance, it should contribute sufficiently novel ideas that are valuable to the field. Unfortunately, the two claimed contributions of this paper fall short in terms of innovation.
> >
> > In conclusion, I’m sorry, but I cannot give a positive evaluation of a paper that I believe has problems in both task setup and novelty. Therefore, I maintain my "borderline reject" rating. That said, this is only my personal opinion. The final decision will be made collectively by all reviewers and the AC. My comments are merely for reference. Authors should not be overly concerned. You are free to contact the AC directly to express your concerns about my review. You can also further emphasize your arguments in support of this paper. However, I encourage you to focus more on producing work with greater value. I wish you all the best in your research endeavors.

---

### Official Review · Reviewer_6UAg · 2024-11-01

**Soundness:** 3
**Presentation:** 3
**Contribution:** 3
**Rating:** 5
**Confidence:** 3

**Summary:**

The paper propose InstructBrush, an inversion-based method for instruction-driven image editing in diffusion models, addressing limitations in handling complex edits. By extracting editing effects from example image pairs, InstructBrush guides novel edits with Attention-based Instruction Optimization and Transformation-oriented Instruction Initialization to enhance inversion and generalization. The paper also introduces TOP-Bench, a benchmark for evaluating instruction-based editing in open scenarios. InstructBrush shows superior performance and semantic alignment in edits, with plans to release the code and benchmark upon acceptance

**Strengths:**

1. The proposed feature similarity-based unique phrase extraction is a simple yet novel approach. The results of the proposed method produce effectively edited images based on instructions.
2. The paper identifies limitations in the evaluation criteria used in conventional image editing and addresses these by proposing a new benchmark, TOP-Bench, to overcome such challenges.

**Weaknesses:**

1. One of the primary contributions, the attention-based instruction module, seems to be a relatively simple technique, without too much novel insight. The concept of using a learnable cross-attention module to optimize features in cross-attention space has previously been introduced[1].
2. The explanation of when the time-aware instruction is effective or ineffective seems insufficient. Additional details on this aspect would enhance clarity.
3. Adding more real-world image tests could improve the robustness of the results. Further comparative experiments with text-guided editing methods would also strengthen the analysis.

[1] Ye et al. Ip-adapter: Text compatible image prompt adapter for text-to-image diffusion models. arXiv preprint arXiv:2308.06721, 2023

**Questions:**

1. Ablation results regarding the degree of truncation would be insightful. To what extent does truncation impact the final results?
2. In certain cases, IP2P+GPT-4 appears to deliver superior results compared to other methods. Further insights into this comparison would be valuable.
3. If unique phrase extraction is conducted effectively, is it possible to sample images based on these conditions? It would be interesting to see how images generated from unique phrases align with actual concepts. If these generated images closely resemble the intended concepts, it would enhance the persuasiveness of the paper’s claims.

---

> ### Author Response · Authors · 2024-11-20
>
> Thank you very much for your constructive comments. We addressed you comments and questions and revised the paper in *brown* color. We summarize the answers below.
>
> **[W1]  The attention-based instruction module, seems to be a relatively simple technique, without too much novel insight.**
>
> We fully understand the reviewer's emphasis on innovation, but as *Reviewer 8Vpr* stated in *Strength 2*, we emphasize that the innovation of our method does not lie in proposing a novel module. Instead, it delves into the extraction of editing concepts based on image prompts. We leverage the prior implementation of Instruction-based image editing models and innovatively introduce the KV token optimization in Attention for image prompts learning to represent the target editing instructions, enhancing the representational power of the concept. Furthermore, to further explore the impact of instructions on editing process, we introduce a time-aware method to optimize the instructions at different denoising times, comprehensively investigate the representation of editing instructions, and provide a clear solution for visual prompt editing.
>
>
>
>
>
> **[W2] The explanation of when the time-aware instruction is effective or ineffective seems insufficient. Additional details on this aspect would enhance clarity**
>
> Thanks for the constructive suggestion. Time-aware optimization method emphasizes learning its focused edits at different denoising time steps. The edits obtained using the time-aware optimization approach are more stable and can be learned in more detail. We have described in detail in *Section 6.2* that this module plays a key role in learning fine-grained edits, and visualize more results on the ablation of the time-aware module in *Figure 12*.
>
>
> **[W3] Adding more real-world image tests could improve the robustness of the results. Further comparative experiments with text-guided editing methods would also strengthen the analysis**
>
> Just as the reviewer is concerned about the application of our method in real image scenarios and its comparison with text-guided methods, we have not overlooked these aspects either. *Figure 11* demonstrates the application of our method on real images, where both local and global edits can extract the editing effects from reference image pairs and apply them to new images, further highlighting the generalization capability of our method. *Figure 13* shows a comparison of results between our method and text-based editing methods. Text-based editing methods have failed in extracting the editing effects between reference image pairs, further demonstrating the significance of our method's design.

---

> ### Author Response · Authors · 2024-11-21
>
> **[Q1]  Ablation results regarding the degree of truncation would be insightful. To what extent does truncation impact the final results?**
>
> Thank you for your clarification. We conduct an ablation study on truncation to explore its contribution to the overall method. As shown in *Figure 14*, we visualized the initialized instructions and editing results with and without truncation (column 5-6). The results indicate that without truncation, incorrect semantic priors are introduced, hindering the learning of the desired editing effects, e.g. edit in the first line to get red glasses.
>
>
>
>
>
>
> **[Q2] In certain cases, IP2P+GPT-4 appears to deliver superior results compared to other methods. Further insights into this comparison would be valuable.**
>
> Thank you for the constructive suggestions. In some instances, the editing results of IP2P outperform Analogist as well as Visii in terms of image quality, and we analyze the reasons as follows, and add it in *line 415* of the paper.
> IP2P+GPT-4 employs text instructions to guide image editing. Although this text-based editing approach cannot accurately extract the editing concepts between image pairs, the generalization ability of text and the model priors of IP2P ensure the quality of the generated images.
>
> In contrast, Analogist leverages the priors of the inpainting diffusion model, and compared to IP2P, it has a lesser understanding of the editing instructions provided by GPT-4. Additionally, the extra structural constraints imposed on the attention further exacerbate its lower adherence to the instructions. For example, suboptimal results are observed in the local edits from row 1 to row 3 in *Figure 3*.
>
> Visii optimizes text tokens to specifically learn the editing concepts between image pairs. However, its initialization method introduces excessive content information unrelated to the edits from the reference image, causing content leakage, such as the leaves in row 2 and the vehicles in the background of the row 3 in *Figure 3*.
>
> In contrast, our method can more accurately extract the editing concepts between reference image pairs, as reflected in our results being more consistent with the ground truth.
>
>
>
> **[Q3] If unique phrase extraction is conducted effectively, is it possible to sample images based on these conditions? It would be interesting to see how images generated from unique phrases align with actual concepts. If these generated images closely resemble the intended concepts, it would enhance the persuasiveness of the paper’s claims.**
>
> Thank you for your thought-provoking questions.
> We believe that relying solely on textual conditions may not fully accomplish this task. For simple image transformations such as cat-to-dog or peach-to-apple, ideal unique phrases can capture the concept to the maximum extent. However, for pixel-level or even mask-level transformations, such as tone editing, different instances may require different transformations at the pixel level. For such tasks, it is challenging to guide editing through textual conditions, so employing subsequent optimization methods to learn these abstract changes is necessary. This also explains why we optimize instructions in the attention feature space rather than the token space of text input, as it can fully represent pixel-level information.
>
> Additionally, the reviewer's suggestion to visualize the editing results of unique phrases is highly constructive. We have consequently shown several sets of transformation effects in *Figure 14*, along with the extracted unique phrases and the results of editing using these unique phrases, to verify the effectiveness of this module.

---

> ### Author Response · Authors · 2024-11-25
>
> Dear reviewer 6UAg,
>
> Given the discussion phase is quickly passing, we want to know if our response resolves your concerns. If you have any further questions, we are more than happy to discuss them. Thanks again for your valuable suggestions!
>
> Best, All anonymous authors

---

> > ### Comment · Reviewer_6UAg · 2024-11-25
> > **Response to authors**
> >
> > Thank you for your thoughtful and constructive responses to my comments.
> >
> > However, despite your detailed replies, I still have some unresolved concerns.
> >
> > The first issue pertains to the novelty of the proposed method. I understand author's point, as well as Reviewer 8Vpr's comments, emphasizing that the paper does not primarily focus on proposing a novel module. However, the method itself constitutes a significant portion of the paper. As highlighted by Reviewer uD16, numerous prior works have already delved deeply into similar methods. This makes it difficult for me to identify the original innovation in this paper. Furthermore, as I mentioned in my previous question, the performance of IP2P+GPT-4 appears to be superior to your proposed method in several results presented in the paper. Considering these points, I find it challenging to discern the advantages the proposed approach.
> >
> > Additionally, I carefully reviewed Reviewer 8Vpr's comment on "A comment on the visual prompt setting" and agree with the practicality and benefits of editing images based on {before, after} image pairs. However, for this argument to be compelling, more experimental results on real-world images should be provided to validate its applicability.
> >
> > Regarding the unique phrase extraction process, I noticed that truncation appears to influence the editing results. However, I believe that when the target phrase to be extracted varies across experiments, the truncation conditions will also need to be adjusted accordingly. Thus, it is necessary to examine how the results change with varying truncation conditions.

---

### Official Review · Reviewer_wgFX · 2024-11-04

**Soundness:** 3
**Presentation:** 3
**Contribution:** 3
**Rating:** 5
**Confidence:** 3

**Summary:**

This paper addresses the challenge of edits that are difficult to describe through user input alone. To address this, it introduces the InstructBrush method, which includes techniques such as Attention-based Instruction Optimization and Transformation-oriented Instruction Initialization. Additionally, it presents an evaluation benchmark, TOP-Bench, to assess the method's performance. Quantitative results demonstrate a competitive performance.

**Strengths:**

The paper highlights an important observation that many practical edits are indeed challenging to articulate in natural language. In response, the authors introduce a benchmark for evaluating performance on this issue, and their quantitative results indicate strong performance.

**Weaknesses:**

The primary concern is that, although the paper aims to address the challenge of edits that are hard to describe in language, the authors still rely on converting image pairs into a descriptive template with a fixed vocabulary list. It would seem more logical to explore the direct relationship between image features and to encode these editing directions directly within the generation model. If transitioning these concepts into language is essential, then the focus should be on translating image differences into descriptive language. Existing editing methods appear sufficient for such language-based tasks.

Additionally, some of the qualitative results highlighted in the paper remain suboptimal, especially when compared to IP2P+GPT-4o.

**Questions:**

1. In line 281, the notation "CAP_x" is used but not previously defined. Is this intended to be "P_x"?
2. The Time-aware Instruction method does not appear effective in the Peach-Apple example in Figure 5. Could an ablation study on this be provided?
3. It would be beneficial to include an ablation study on the optimization of the initial m tokens compared to optimizing the full set of tokens.

---

> ### Author Response · Authors · 2024-11-20
>
> Thank you very much for your constructive comments. We addressed you comments and questions and revised the paper in blue color. We summarize the answers below.
>
> **[W1]  [...] more logical to explore the direct relationship between image features and to encode these editing directions directly [...], the focus should be on translating image differences into descriptive language [...], Existing editing methods appear sufficient for such language-based tasks**
>
> 1. Our introduction of image pairs to guide editing does not depend entirely on converting image pairs to text. We emphasize that the essence of our method is based on optimization. Our method compares the differences between images pairs through the diffusion process to optimize the editing direction in the instruction space. The introduction of unique phrases only adds semantic information to the optimization process (as shown in Table 2, where the initialization module's impact on the semantic score CLIP-D). Similar methods have also been implemented in [1] and [2]. The method of encoding the differences between image pairs as a signal guided diffusion model is essentially an encoder based method and is not within the scope of our research.
>
> 2. We would like to clarify that we indeed only convert the differences between images into natural language to introduce semantic priors related to editing for the optimization. To this end, we have designed the "Transofmation-oriented Instruction Initialization" method, which extracts unique phrases representing the differences between pre- and post-edited images as semantic priors to assist in subsequent instruction optimization.
>
> 3. We reiterate that what we aim to solve is not text-based editing tasks, but rather to precisely replicate the editing effects between image pairs. The conversion of natural text related to editing cannot achieve this function, but it can serve as semantic priors to aid in the subsequent optimization process.
>
> [1] Visual Instruction Inversion: Image Editing via Visual Prompting, NIPS 2023
>
> [2] Analogist: Out-of-the-box Visual In-Context Learning with Image Diffusion Model, Siggraph 2024
>
>
>
>
> **[W2] Some of the qualitative results highlighted in the paper remain suboptimal, especially when compared to IP2P+GPT-4o**
>
> We fully understand the concerns of the reviewer, and a good visual result is not only pleasing but also helps to understand the advantages of the method. Compared with IP2P+GPT-4o, our method achieves more consistent editing results with the reference image pairs. As we demonstrated in *Figure 1* in the paper, in the first row, we change the entire tone of the input image to be consistent with the reference image. However, IP2P+GPT-4o cannot control the pixel level and even mask level transformations with reference examples. In the second row, the smile of our method is more consistent with the reference image pairs, and although IP2P+GPT-4o also captures this concept, there is a significant difference in the degree of representation compared to the reference image. There are still many similar examples in *Figure 3*, indicating that compared to IP2P+GPT-4o, our method is more capable of replicating the editing effects of example image pairs.
>
> We fully appreciate the concerns of the reviewer and hope to be provided examples of what the reviewer consider to be suboptimal results, in order to update these results in the latest version and present better visual effects to readers.

---

> ### Author Response · Authors · 2024-11-20
>
> **[Q1] The notation "CAP_x" is used but not previously defined.**
>
> Thank you for the careful correction. Exactly as the reviewer's judgment, we have corrected the typo error of CAP_x in line 281 as P_x.
>
>
>
> **[Q2] The Time-aware Instruction method does not appear effective in the Peach-Apple example in Figure 5. Could an ablation study on this be provided?**
>
> Thank you for your positive feedback. We believe that the reason why time-aware module did not significantly improve the result of the peach-apple editing in the second row of *Figure 5* is that this type of editing is relatively simple. Using other modules are sufficient to achieve this editing effect. The fine-grained face glasses editing in the first row of *Figure 5* proves the significance of the time-aware module setting. In addition, we provide quantitative results of the ablation experiment in *Table 2*, the improvement of the metrics indicates the necessity of the module design.
> In addition, we provide more qualitative results on the ablation of the time-aware module in *Figure 12* of the paper to visualize the effectiveness of this design.
>
>
> **[Q3] It would be beneficial to include an ablation study on the optimization of the initial m tokens compared to optimizing the full set of tokens.**
>
> Thank you for your constructive suggestions. We are indeed optimizing the initial m tokens. Because we believe that optimizing all tokens is not conducive to the generalization of the editing instructions to be extracted. We have ablated the impact of optimizing the first m tokens initialized in KV and the impact of optimizing all tokens in Table 4 of the paper. The results show that optimizing the first m tokens, which is our current setting, yields better effects.
>
> **Table 4: Extra Ablation Study. We ablate the impact of optimizing the first $m$ tokens initialized in K, V (Ours) and the impact of optimizing all tokens.**
>
> | Setting     | PSNR ↑ | SSIM ↑  | LPIPS ↓ | CLIP-D ↓ | CLIP-I ↑  | DINO ↑  |
> |-------------|--------|---------|---------|----------|-----------|---------|
> | All tokens  | 19.66  | 0.6660  | 0.2254  | 0.3214   | **0.9379**| 0.9114  |
> | **Ours**    | **20.68** | **0.6922** | **0.1918** | **0.3140** | 0.9115   | **0.9412** |

---

> ### Author Response · Authors · 2024-11-25
>
> Dear reviewer wgFX,
>
> Given the discussion phase is quickly passing, we want to know if our response resolves your concerns. If you have any further questions, we are more than happy to discuss them. Thanks again for your valuable suggestions!
>
> Best, All anonymous authors

---

> ### Comment · Reviewer_wgFX · 2024-11-27
>
> Thank you for your responses. I have carefully reviewed all the comments and your replies. Some of my concerns have been addressed. However, as reviewers 6UAg and uD16 have also noted, I remain worried about the technical novelty and performance superiority of this paper. Therefore, I maintain my current score.

---

### Official Review · Reviewer_8Vpr · 2024-11-04

**Soundness:** 3
**Presentation:** 3
**Contribution:** 2
**Rating:** 6
**Confidence:** 4

**Summary:**

This paper studies visual prompting for image editing settings, in which the edit is given by a {before, after} pair. Building upon existing works, this paper aims to improve current approaches by adding more detail and precision to the editing (e.g., fine-grained details, localization). To that end, a novel framework called InstructBrush is introduced, which includes Attention-based Instruction Optimization and Transformation-oriented Instruction Initialization. Quantitative and qualitative results indicate the effectiveness of the proposed framework.

**Strengths:**

- This paper is among the first works focusing on more fine-grained details in visual prompting for image editing. Most existing works focus on learning the "global" edit from a given {before, after} pair (e.g., turning a photo into a watercolor style), which often fails to capture fine-grained details (e.g., editing only the color of the water in a cup).
- While "attention manipulation" for image editing is not new, using this technique in "visual prompting" is novel and interesting.
- The experiments section is extensive and results are convincing.

**Weaknesses:**

Please make sure to attribute the correct papers. There are some formulas in the manuscript that do not seem to be correctly attributed to the original papers. For example, in line 760, I believe this formula was proposed or partially proposed in [1, 2, 3, etc.].

References:
1. Zero-shot Image-to-Image Translation (SIGGRAPH 2023)
2. Visual Instruction Inversion: Image Editing via Visual Prompting (NeurIPS 2023)
3. StyleGAN-NADA: CLIP-Guided Domain Adaptation of Image Generators (SIGGRAPH 2022)

**Questions:**

Generally, I am inclined to accept this paper. At this time, I do not have any major concerns regarding its technical aspects or novelty. My only concerns are related to attribution to previous works, as there are instances where it may come across as overclaiming. I'd rate this paper 6.5-7, but unfortunately this there is no 7 option in the rating.

---

> ### Author Response · Authors · 2024-11-21
>
> Thank you to the reviewer for the positive evaluation and valuable feedback on our work. Firstly, we are honored to have made some progress in the fine-grained editing for visual prompting, and we appreciate your recognition of our efforts.
>
> At the same time, we take the issue of proper citation very seriously and are willing to detail in our response how we plan to improve this aspect. We have conducted a thorough review and verification of this part and found that there are indeed some formulas that were not properly attributed, particularly the formula mentioned on line 760. We have now cited these references to enhance the transparency and academic integrity of our work, which also helps readers better understand the origin and development of these formulas.
>
> We fully understand the importance of citation accuracy in academic research and are committed to strictly adhering to this standard in our revised manuscript. Thank you once again for your review and suggestions, which are crucial for enhancing the quality of our work. If you have any further suggestions or need additional clarification, please feel free to let us know, as we are more than willing to discuss further.

---

### Meta-Review · Area_Chair_m1KZ · 2024-12-24

**Metareview:**

The paper proposes InstructBrush, a method aimed at improving the quality of image editing techniques. The method introduces novel instruction optimization and initialization techniques to improve editing fidelity and spatial precision. Experimental results show improved performance across multiple benchmarks compared to existing methods. Key strengths include the method's ability to handle edits that are hard to describe textually and its overall practicality for real-world editing tasks. However, reviewers pointed out limited technical novelty compared to existing prompt-based image editing methods and questioned whether similar results could be achieved with text-based approaches alone. Additional weaknesses include inadequate evaluation metrics and unclear performance comparisons with current state-of-the-art methods.

While the paper does demonstrate results on an interesting new task, reviewers deem this draft to need a bit more work before publication. It's advised that the authors revise and resubmit.

**Additional Comments On Reviewer Discussion:**

Reviewers raised concerns about technical novelty, presentation clarity, and insufficient evaluation against state-of-the-art methods like GPT-4V. The authors responded by clarifying their novel contributions in the attention-based instruction optimization and providing additional benchmark results, but borderline negative reviews remained negative, citing concerns about novelty and quality.

---

### Decision · Program_Chairs · 2025-01-22

Reject